# Characteristics of intercontinental transport of tropospheric ozone from Africa to Asia

Han Han[1], Jane Liu[1,2], Huiling Yuan[1], Bingliang Zhuang[1], Ye Zhu[1,3], Yue Wu[1], Yuhan Yan[4], Aijun Ding[1]

[1]School of Atmospheric Sciences, Nanjing University, Nanjing, China

[2]Department of Geography and Planning, University of Toronto, Toronto, Canada

[3]Shanghai Public Meteorological Service Centre, Shanghai, China

[4]Chinese Academy of Science, Institute of Atmospheric Physics, Beijing, China

*Correspondence to:* Jane Liu ([janejj.liu@utoronto.ca](janejj.liu@utoronto.ca))

**Abstract**

In this study, we characterize the transport of ozone from Africa to Asia through the analysis of the simulations of a global chemical transport model, GEOS-Chem, from 1987 to 2006. The receptor region Asia is defined within $5^oN$-$60^oN$ and $60^oE$-$145^oE$, while the source region Africa is within $35^oS$-$15^oN$ and $20^oW$-$55^oE$ and within $15^oN$-$35^oN$, $20^oW$-$30^oE$. The ozone generated in the African troposphere from both natural and anthropogenic sources is tracked through tagged ozone simulation. Combining with analysis of trajectory simulations using the Hybrid Single-Particle Lagrangian Integrated Trajectory (HYSPLIT) model, we find that the upper branch of the Hadley cell connects with the subtropical westerlies in the northern hemisphere (NH) to form a primary transport pathway from Africa to Asia in the middle and upper troposphere throughout the year. The Somali jet that runs from eastern Africa near the equator to the Indian subcontinent in the lower troposphere is the second pathway that appears only in NH summer.

The influence of African ozone mainly appears over Asia south of 40°N. The influence shows strong seasonality, varying with latitude, longitude, and altitude. In the Asian upper troposphere, imported African ozone is largest from March to May around 30°N (12-16 ppbv) and lowest during July-October

around 10°N (~2 ppbv). In the Asian middle and lower troposphere, imported African ozone peaks in NH winter between 20-25°N. Over 5-40°N, the mean fractional contribution of imported African ozone to the overall ozone concentrations in Asia is largest during NH winter in the middle troposphere (~18%) and lowest in NH summer throughout the tropospheric column (~6%).

This seasonality mainly results from the collective effects of the ozone precursor emissions in Africa, and meteorology and chemistry in Africa and Asia and along the transport pathways. The seasonal swing of the Hadley circulation and subtropical westerlies along the primary transport pathway plays a dominant role in modulating the seasonality. There is more imported African ozone in the Asian upper troposphere in NH spring than in winter. This is likely due to more ozone in the NH African upper troposphere from biogenic and lightning $NO_x$ emissions in NH spring. The influence of African ozone

on Asia appears larger in NH spring than in autumn. This can be attributed to higher altitudes of the elevated ozone in Africa and stronger subtropical westerlies in NH spring. In NH summer, African ozone hardly reaches Asia because of the blockings of Saharan High, Arabian High, and Tibetan High on the transport pathway in the middle and upper troposphere, in addition to the northward swing of the subtropical westerlies. The seasonal swings of the Intertropical Convergence Zone (ITCZ) in Africa,

coinciding with the geographic variations of the emissions, can further modulate the seasonality of the transport of African ozone, owing to the functions of the ITCZ in enhancing lightning $NO_x$ generation and uplifting ozone and ozone precursors to upper layers. The strength of the ITCZ in Africa is also found to be positively correlated with the interannual variation of the transport of African ozone to Asia in NH winter.

Ozone from NH Africa makes up over 80% of the total imported African ozone over Asia in most altitudes and seasons. The interhemispheric transport of ozone from the southern hemispheric Africa is most evident in NH winter over the Asian upper troposphere and in NH summer over the Asian lower troposphere. The former case is associated with the primary transport pathway in NH winter, while the

latter case is with the second transport pathway. The intensities of the ITCZ in Africa and the Somali jet

can respectively explain ~30% of the interannual variations in the transport of ozone from the southern

hemispheric Africa to Asia in the two cases.

## 1 Introduction

Tropospheric ozone is a major air pollutant, harmful to human health (Anenberg et al., 2010),

agricultural crops, and natural ecosystems (Hollaway et al., 2012; Lefohn et al., 2017). It also acts as a

greenhouse gas, whose global mean radiative forcing is about $0.4 \pm 0.2$ W/m$^2$ for the period 1750-2011

(Myhre et al., 2013). The dominant source of tropospheric ozone is from the photochemical reactions in

which volatile organic compounds (VOCs) and carbon monoxide (CO) are oxidized in the presence of

nitrogen oxides (NO$_x$) (Lelieveld and Dentener, 2000). Transport of ozone from the stratosphere

downward is another source of tropospheric ozone (Neu et al., 2014; Akritidis et al., 2016). Due to its

lifetime of days to weeks in the free troposphere, ozone can be transported over long distances across

continents, as shown in a wealth of observation evidence (Huntrieser et al., 2005; Lewis et al., 2007;

Verstraeten et al., 2015). Consequently, tropospheric ozone in a region is greatly influenced by ozone

transport from upwind regions (Doherty et al., 2013, 2017). Therefore, intercontinental transport of

ozone has been a significant issue to scientists, public, and policymakers concerned with air quality and

climate change (HTAP, 2010; Cooper et al., 2015; Doherty, 2015; Huang et al., 2017).

There have been numerous studies on the major source-receptor relationships in the northern

hemisphere (NH) for ozone transport among Asia, North America, and Europe, which are trans-Pacific

(Jaffe et al., 2003; Zhao et al., 2003; Cooper et al., 2010; Nopmongcol et al., 2017), trans-Atlantic (Wild

et al., 1996; Cooper et al., 2002; Guerova et al., 2006; Derwent et al., 2015; Karamchandani et al., 2017;

Knowland et al., 2017), and trans-Eurasian (Wild et al., 2004; Fiore et al., 2009; Li et al., 2014b)

transport. The trans-Pacific transport from Asia to North America has been studied most because of the

efficient eastward export of high pollutants in Asia and the close connection to legislative issues related to the control of ozone in USA (Cooper et al., 2010, 2011, 2015; Verstraeten et al., 2015; Huang et al.,

2017; Lin et al., 2017). Developing a better understanding of the intercontinental transport of ozone across the NH is one of the main objectives for the Task Force on Hemispheric Transport of Air Pollution (TF HTAP) (Galmarini et al., 2017, http://www.htap.org).

Transport of ozone to Asia from various source regions in the world has been less documented but received increasing attention since the 2000s (Holloway et al., 2008; Nagashima et al., 2010; Wang et

al., 2011; Li et al., 2014b; Chakraborty et al., 2015; Zhu et al., 2016; Zhu et al., 2017b). The distribution of foreign ozone in Asia is modulated by numerous processes, involving chemistry and meteorology in both Asia and the various source regions, as well as along the transport pathways (Wild et al., 2004; Li et al., 2014b). Previous studies illustrated the important role that meteorology plays in these processes (Sudo and Akimoto, 2007; Sekiya and Sudo, 2012, 2014; Zhu et al., 2017b). For example, governed by

the westerlies in the NH, the influence of European and North American ozone on Asia is larger in higher latitudes than in lower altitudes (Wild et al., 2004; Holloway et al., 2008; Zhu et al., 2017b). The seasonal switch in Asian monsoonal winds significantly affects the seasonal variation of ozone transported to Asia (Nagashima et al., 2010; Wang et al., 2011; Zhu et al., 2017b). The Asian monsoon anticyclone (South Asian High, SAH) can also modulate the movement of ozone in the upper

troposphere over Eurasia (Vogel et al., 2014; Garny and Randel, 2016).

Africa covers areas from the NH to the Southern Hemisphere (SH), with around three quarters of the continent in the tropics. Comparing with other continents, Africa has the most frequent lightning (Albrecht et al., 2016) and the largest burned areas (Giglio et al., 2013). Approximately, 70% of tropospheric ozone produced over the African troposphere is exported out of Africa (Aghedo et al.,

2007), making African ozone an obvious influence on the global tropospheric ozone budget (Piketh and Walton, 2004; Thompson, 2004; Williams et al., 2009; Bouarar et al., 2011; Zare et al., 2014).

Nevertheless, there is relatively little literature on the transport of African ozone to Asia (Liu et al., 2002; Sudo and Akimoto, 2007; Lal et al., 2014). By tagging ozone from Africa in a global chemistry transport model, Liu et al. (2002) reported that African ozone contributed about 10-20 ppbv to the

overall ozone concentrations in the middle and upper troposphere at Hong Kong, China, during the November-April period. Sudo and Akimoto (2007) showed that there is a significant interhemispheric ozone transport from South America to Japan in the upper troposphere in conjunction with ozone export from northern Africa. Lal et al. (2014) reported that the ozone distribution over western India in the lower troposphere during the summer monsoon season is affected by long-range transport from the east

coast of Africa. These previous studies mostly focused on the transport of African ozone to some locations in Asia. It is unclear to which degree that African ozone influences Asia regionally, how the transport pathways vary with season, and what underlying mechanisms modulate the transport.

The Intertropical Convergence Zone (ITCZ), defined as the convergence of the trade winds, is one of the most prominent meteorological phenomena in the tropics (Waliser and Gautier, 1993; Žagar et al.,

2011; Suzuki, 2011). ITCZ is a heat engine driving the ascending branch of the Hadley circulation (Nicholson, 2009; Suzuki, 2011). ITCZ and its seasonality can significantly impact the meteorological conditions over Africa (Nicholson, 2009; Collier and Hughes, 2011; Suzuki, 2011). Characterized with deep and strong convection, the equatorial region between the ITCZ branches is a region with effective interhemispheric mixing (Avery et al., 2001). The convective divergence in the upper troposphere over

the ITCZ was proposed to be one of the primary mechanisms for the interhemispheric transport (Hartley and Black, 1995; Avery et al., 2001). Meanwhile, the ITCZ can be a barrier to interhemispheric exchange in the lower troposphere (Raper et al., 2001). Together with the Southeast Asian monsoon, the movement of the ITCZ can modulate pollution transport to Southeast Asia (Pochanart et al., 2003, 2004). In addition, the ITCZ over western Africa is found to play a significant role in controlling the

transport of Saharan mineral dust to northern South America, modifying air quality and climate there

(Piketh and Walton, 2004; Doherty et al., 2012, 2014; Rodríguez et al., 2015). However, the influence of the ITCZ on ozone transport from Africa to Asia has not been well understood.

In this paper, we present a comprehensive study on the transport of African ozone to Asia through analysis of tagged ozone simulations, as well as trajectory simulations. In this paper, the term "African ozone" refers to ozone that is generated in the African troposphere below the tropopause from both natural and anthropogenic sources, as Sudo and Akimoto (2007) suggested that investigating the contributions of both sources to ozone production is necessary. The term "imported African ozone" or "imported ozone" refers to the African ozone that has been transported to and distributed over Asia, and it refers to African ozone concentrations in the model grids in Asia. The source region Africa and the receptor region Asia are shown in Fig. 1. The contribution of the source regions to the receptor region is presented as absolute or fractional contribution. The former refers to the imported African ozone concentrations in ppbv, while the latter is the ratio of the imported ozone concentrations to the overall ozone concentrations in the model grids in Asia. Our specific objectives are (1) to characterize the seasonal variations of ozone transport from Africa to Asia, (2) to investigate the underlying mechanisms responsible for such seasonal variations, and (3) to find meteorological influences, including the ITCZ, on the interannual variation of the transport of African ozone to Asia, including the interhemispheric transport from the SH. Our analysis is based on the simulations from a 3-dimensional global chemical transport model, GEOS-Chem (Bey et al., 2001a), for 20 years from 1987 to 2006, supplemented by the analysis of trajectory simulations from the Hybrid Single-Particle Lagrangian Integrated Trajectory model (HYSPLIT) (Draxler and Hess, 1998; Stein et al., 2015). In section 2, GEOS-Chem and the tagged ozone stimulation method are described, followed by the comparison of GEOS-Chem simulations with ozonesonde and satellite data. The HYSPLIT model and the meteorological data are also described. Section 3 is the core section that discusses the seasonality of imported African ozone over Asia and possible underlying mechanisms.  In section 4, meteorological influences on the

interannual variation in the transport of African ozone to Asia, including the interhemispheric transport,

are explored. Finally, conclusions are drawn with discussion in section 5.

**2 Models and data**

**2.1 The description of the GEOS-Chem model**

A global 3-dimensional chemistry transport model, GEOS-Chem (version v9-02) (Bey et al., 2001a,

http://geos-chem.org), is employed to simulate the global ozone distributions and the source-receptor

relationship between Africa and Asia. GEOS-Chem includes a detailed description of tropospheric $O_3$-

$NO_x$-hydrocarbon-aerosol. It is driven by assimilated meteorological data from the GEOS at NASA

Global Modeling and Assimilation Office (GMAO).

In this study, the simulations are driven by GEOS-4 meteorology at a $4^o$ latitude by $5^o$ longitude

horizontal resolution, degraded from their native resolution of 1° latitude by 1.25° longitude. There are

30 vertical layers including 17 levels in the troposphere. Transport and chemical timesteps are 10

minutes and 20 minutes, respectively, as suggested by Philip et al. (2016). GEOS-4 uses the deep

convection scheme of Zhang and McFarlane (1995) and the shallow convection scheme of Hack (1994).

By comparing the GEOS-Chem simulations driven by GEOS-4 and GEOS-5 with satellite observations

in the tropical troposphere, Liu et al. (2010) and Zhang et al. (2011) have found GEOS-4 has stronger

deep convection in the tropics than GEOS-5. In general, GEOS-Chem driven by GEOS-4 can simulate

tropospheric ozone concentrations that are in good agreement with ozonesonde observations (Liu et al.,

2010; Zhang et al., 2011; Choi et al., 2017; Zhu et al., 2017b).

GEOS-Chem has various modes that serve different simulation goals. In the full chemistry

simulation, the emission inputs for a specific year are scaled from each inventory's base year. The

global anthropogenic emission inventories are from EDGAR 3.2 for $NO_x$, CO, $SO_x$ (Olivier and

Berdowski, 2001) and RETRO for VOC emissions (Pulles et al., 2007) in 2000, merged with the

following regional inventories: the INTEX-B Asia emissions inventory in 2006 (Zhang et al., 2009), the

US Environmental Protection Agency National Emission Inventory in 2005 (NEI05) for North America, the Cooperative Programme for Monitoring and Evaluation of the Long-range Transmission of Air Pollutants in Europe (EMEP) inventory for Europe (Vestreng and Klein, 2002) in 2005, Big Bend Regional Aerosol and Visibility Observational (BRAVO) Study Emissions Inventory for 1999 in Mexico (Kuhns et al., 2003), and the Criteria Air Contaminants (CAC) inventory for Canada in 2005.

Biofuel emissions are from Yevich and Logan (2003). Biomass burning and biogenic emissions are from the GFED3 inventory (van der Werf et al., 2010) and MEGAN 2.1 (Guenther et al., 2012) respectively. Lightning $NO_x$ emissions are calculated with the scheme of Allen et al. (2010) and the vertical distribution suggested by Ott et al. (2010). The annual global lightning $NO_x$ emissions is 5.97 Tg N yr$^{-1}$, comparable to 6±2 Tg N yr$^{-1}$ in Martin et al. (2007) and 6.3 Tg N yr$^{-1}$ in Miyazaki et al. (2014). The annual total lightning $NO_x$ emission in Africa is 1.6 Tg N yr$^{-1}$, 0.80 Tg N yr$^{-1}$in NHAF, and 0.79 Tg N yr$^{-1}$ in SHAF (see supplementary Figs. S1and S2). The anthropogenic CO and $NO_x$ emissions from GEOS-Chem for 2000 are compared with those in the HTAP2 emission inventories for 2008 (http://edgar.jrc.ec.europa.eu/htap_v2/) (Figs. S3 and S4). The annual anthropogenic CO emissions in Africa are 12.2 Tg yr$^{-1}$ for 2000 in GEOS-Chem, lower than 62.5 Tg yr$^{-1}$ for 2008 from the HTAP2 inventories. The anthropogenic NOx emissions in GEOS-Chem are 2.27 Tg yr$^{-1}$ for 2000, also lower than 4.53 Tg yr$^{-1}$for 2008 from the HTAP2 inventories. Although the anthropogenic emissions contribute less significantly to the ozone generation in Africa than biogenic, biomass burning, and lightning emissions (Aghedo et al., 2007), the differences between these emission inventories imply that African ozone simulated by GEOS-Chem is with some uncertainties.

To track the transport of ozone generated in Africa, we use the tagged ozone mode in GEOS-Chem, in which odd oxygen is tagged ($O_x = O_3 + NO_2 + 2NO_3 + 3N_2O_5 + HNO_3 + HNO_4 + PAN + PMN + PPN$, Fiore et al., 2002; Zhang et al., 2008). Since ozone accounts for most of $O_x$, we refer to ozone instead of

$O_x$ for clarity. To prepare for the tagged ozone simulation, we first run GEOS-Chem in the full chemistry mode to generate the daily ozone production rate and loss frequency. Then we run GEOS-Chem in the tagged ozone mode to differentiate ozone produced in different source regions, tagged as different tracers, by using the archived daily ozone production rates and loss frequencies. As shown in Fig. 1, the source region, Africa, is defined as the region of $35^{o}S-15^{o}N$, $20^{o}W-55^{o}E$ and $15^{o}N-35^{o}N$, $20^{o}W-30^{o}E$. Therefore, ozone produced in Africa below tropopause from all natural and anthropogenic sources is tagged as a tracer. We further divide Africa into northern hemispheric Africa (NHAF) and southern hemispheric Africa (SHAF), which is further separated by the tropospheric layers, the lower troposphere (LT, from the surface to 700 hPa), the middle troposphere (MT, 700-300 hPa), and the upper troposphere (UT, 300 hPa-tropopause) so to add six more tracers. The receptor region, Asia, is defined as the region of $5^{o}N-60^{o}N$, $60^{o}E-145^{o}E$.

In the simulation for a year, both seasonal variation of chemistry and meteorology are considered. To study the meteorological influence on the interannual variation in the transport of African ozone to Asia, the tagged ozone simulation is conducted for 20 years from January 1986 to December 2006 (using 1986 for model spin-up). As our purpose is focused on the meteorological influence on transport, we allow the meteorology to vary from year to year but fix the chemistry constantly, i.e., using the archived daily ozone production and loss data in one year, i. e. 2005, for the 20 year simulation. This treatment is similar to previous studies (Liu et al., 2005; Sekiya and Sudo, 2014). Note that meteorology affects both transport (a physical process) and chemistry while this treatment only considers the former.

Numerical approaches to exploring the source-receptor relationships include (1) tagged tracer simulations, (2) trajectory simulations, (3) perturbation simulations, and (4) inverse simulations (Zhu et al., 2017b). Compared with the other approaches, the tagged trace simulation can track ozone effectively from different source regions and to separate the contributions of ozone from different source regions to a receptor region and quantify each region's contribution. One issue with the tagged

ozone simulation is the nonlinearity of chemistry. This nonlinearity can cause large differences between the full chemistry and the tagged ozone simulations. To test the scale of the nonlinearity in the simulations, ozone concentrations between the full chemistry and the tagged ozone simulations are compared at four ozonesonde sites in Africa and India (Fig. S5). The difference is found to generally within ±5%, suggesting that this approach works in these regions without large bias.

## 2.2 The validation of GEOS-Chem simulations

GEOS-Chem has been used widely in studying pollution transport (Bey et al., 2001b; Koumoutsaris et al., 2008; Zhang et al., 2008; Liu et al., 2011; Wang et al., 2011; Ridder et al., 2012; Long et al., 2015; Jiang et al., 2016; Huang et al., 2017; Ikeda et al., 2017; Zhu et al., 2017a; Zhu et al., 2017b). Tropospheric ozone simulated by GEOS-Chem has been extensively validated using ozonesonde and satellite data, such as in North America (Zhang et al., 2008; Zhu et al., 2017b), Europe (Kim et al., 2015), East Asia (Wang et al., 2011; Zhu et al., 2017a; Zhu et al., 2017b), and other regions (Liu et al., 2009). These validation practices have suggested reasonable agreements between the model simulations and ozone measurements. In this study, for an enhanced confidence on the model performance, we compare the GEOS-Chem simulations with the ozonesonde data in Africa and India and with the satellite measurements from the Tropospheric Emission Spectrometer (TES) instrument. The ozonesonde data are acquired from the World Ozone and Ultraviolet Radiation Data Centre (WOUDC) (http://www.woudc.org/home.php) and the monthly TES product TL2O3LN is from the NASA Langley Atmospheric Science Data Center (https://eosweb.larc.nasa.gov/project/tes/tes_table).

Three ozonesonde stations in Africa are selected for their long record. The stations include Santa Cruz in North Africa (28.42$^o$N, 16.26$^o$W, 36 m), Nairobi in East Africa (1.27$^o$S, 36.8$^o$E, 1745 m), and Irene in South Africa (25.91$^o$S, 28.21$^o$E, 1524 m) (Fig. 1). These ozonesonde data have been used in studying tropospheric ozone in Africa widely (Clain et al., 2009; Thompson et al., 2012, 2014). In Asia,

comparisons between GEOS-Chem simulations and ozonsonde observations have been made by Liu et al. (2002) and Zhu et al. (2017b) at stations over the Pacific Rim, showing that GEOS-Chem can generally capture the vertical and seasonal variations of ozone concentrations in the region. We further compare the GEOS-Chem simulations at three Indian sites, including New Delhi in northern India (28.3$^o$N, 77.1$^o$E, 273 m), Poona in western India (18.53$^o$N, 73.85$^o$E, 559 m), and Thiruvananthapuram in southern India (8.48$^o$N, 76.95$^o$E, 60 m) (Fig. 1).

Fig. 2 shows the simulated and measured vertical ozone profiles by season, which are averaged from 1999 to 2003 at Santa Cruz, from 2003 to 2006 at Nairobi, and from 1999 to 2005 at Irene. Fig. 3 compares the time series of monthly ozone concentrations between the model simulations and ozonesonde measurements at different tropospheric layers at the sites. The corresponding bias, root-mean-square error (RMSE), and the correlation coefficient ($r$) between the two datasets are shown in Table 1 for the vertical profiles and in Table 2 for the time series, respectively.

For the ozone vertical profiles (Fig. 2 and Table 1), GEOS-Chem appears to reasonably capture the ozone vertical variation and its seasonality at the three sites. It appears that GEOS-Chem overestimates ozone in the upper troposphere at Santa Cruz in NH winter and spring and underestimates ozone in the upper troposphere at Nairobi in NH summer and autumn. The correlation coefficients between the two data are above 0.9 for most seasons at the three sites. The mean biases ranges from -9.3% (at Nairobi in October) to 23.2% (at Santa Cruz in January), while the RMSE ranges from 4.3 ppbv (at Nairobi in April) to 15.5 ppbv (at Santa Cruz in January). The bias and RMSE suggest that the model performs no better at a station or a season than the other stations or seasons.

For the time series of ozone at the upper, middle and lower troposphere, as well as the surface layer (Fig. 3 and Table 2), GEOS-Chem also performs reasonably well. The correlation coefficients between the two datasets in the tropospheric layers range within 0.57-0.79 at Santa Cruz, 0.76-0.90 at Nairobi, and 0.61-0.82 at Irene, all significant at a 95% significant level. The correlation coefficient is 0.60, 0.82,

and 0.39 for the surface layer at Santa Cruz, Nairobi, and Irene, respectively. However, the model underestimates upper tropospheric ozone in Nairobi (Table 2, Bias= -14.2%), somewhat overestimate ozone in 500- 300 hPa at Santa Cruz (Table 2, Bias= 4.5%) and near the surface at Irene (Table 2, Bias=60.3%).

The comparison between the GEOS-Chem and ozonesonde data at the three India sites is shown in a
seasonal-altitude distribution in Fig. 4. The ozone concentrations are the means between 1994 and 2003. The time series of ozonesonde data at the sites are not shown because of inadequate records.  The seasonal-altitude patterns of ozone at the three sites are well simulated, although the model overestimates the ozone near the surface and underestimates the ozone in the middle and upper troposphere at the three sites. The annual mean bias at New Delhi is 5.6%, -14.5%, and -18.9% in the
lower (LT), middle (MT), and upper (UT) troposphere, respectively. At Poona, it is 2.1%, -5.6%, and  -19.1% in the LT, MT, and UT, respectively, while the annual mean bias at Thiruvananthapuram is -4.4% in the LT, -13.3% in the MT, and -27.4% in the UT.

The global distribution of ozone from the GEOS-Chem simulations is compared with the TES observation in 2005 in the four seasons at 464 hPa (Fig. S6), a layer around which the TES satellite data
have the least bias. The GEOS-Chem simulations are smoothed with the TES averaging kernels and *a priori*. The smoothed GEOS-Chem simulations resemble the GEOS-Chem simulations (not shown, see Jiang et al. 2016). Generally, GEOS-Chem can capture the global variation of ozone in space and by season, such as the elevated ozone concentrations over the NH middle latitudes in NH spring and summer and the plumes of elevated ozone from biomass burning over southern Africa in NH autumn.
The smoothed ozone concentrations are generally lower than the TES observation over Africa in all the seasons, with a maximum difference of 20 ppbv (or 20%). Note that TES ozone retrievals generally are higher than ozonesonde data, having a mean positive bias of 3-11 ppbv in the troposphere (Nassar et al., 2008).

Overall, GEOS-Chem can generally capture the seasonality of the ozone profile in the ozonsonde

data over Africa. No systematic bias is suggested for a station, a season, or a tropospheric layer. In Asia, GEOS-Chem tends to overestimate ozone in the lower troposphere and underestimate ozone in the upper troposphere at three Indian sites. The GEOS-Chem simulations also compare reasonably with satellite TES data in space and time.

**2.3 The HYSPLIT trajectory model and meteorological data**

To supplement the analysis on the GEOS-Chem simulations, we use the Hybrid Single-Particle Lagrangian Integrated Trajectory model, version 4 (HYSPLIT-4) (Draxler and Hess, 1998; Stein et al., 2015) to simulate forward trajectories and examine the transport pathways of African ozone to Asia. The HYSPLIT model is one of the most extensively used atmospheric transport and dispersion models

(Fleming et al., 2012). Meteorological inputs to HYSPLIT are the NCEP reanalysis at a resolution of $2.5^o \times 2.5^o$ (http://ready.arl.noaa.gov/archives.php). Six-day forward trajectories are calculated from 1987 to 2006 four times a day (00, 06, 12 and 18 UTC) at seven African sites, including Cairo ($31.1^o$E, $30^o$N), Ghat ($10.2^o$E, $24.6^o$N), Khartoum ($32.3^o$E, $15.4^o$N), Abuja ($7.3^o$E, $9^o$N), and Juba ($31.4^o$E, $4.5^o$N) in NHAF and Dar es Salaam ($39.2^o$E, $6.5^o$S) and Luanda ($13.1^o$E, $8.5^o$S) in SHAF (Fig. 1). The levels at

1.5 km, 5.5 km, and 11 km are selected to represent the trajector starting altitude at the lower, middle, and the upper troposphere, respectively. The seven sites are chosen to represent two longitudinal and 3-4 latitudinal zones in Africa.

The meteorological data include the NCEP/NCAR reanalysis I (Kalnay et al., 1996). The daily wind fields are used to describe the climatology of atmospheric circulation during the study period from 1987

to 2006. The product is available on a $2.5^o \times 2.5^o$ horizontal grid at 17 pressure levels from 1000 hPa to 10 hPa (https://www.esrl.noaa.gov/psd/data/gridded/data.ncep.reanalysis.html).

Additionally, the monthly Outgoing Longwave Radiation (OLR) data from NCAR at $2.5^o \times 2.5^o$ with

temporal interpolation (Liebmann and Smith, 1996, https://www.esrl.noaa.gov/psd/data/gridded/
data.interp_OLR .html) are used to indicate the intensity of the ITCZ. The dataset has been widely used

for tropical studies on deep convection and rainfall (Mounier and Janicot, 2004).

## 3 Seasonal variations of the transport of African ozone to Asia

### 3.1 Seasonal variations in African ozone over Asia

Because a substantial amount of ozone and ozone precursors are produced or emitted in Africa,

tropospheric ozone in the other continents is largely influenced by ozone outflow from Africa (Aghedo
et al., 2007; Zare et al., 2014). Fig. 5 shows seasonal variations of imported African ozone in the Asian
troposphere, varying with latitude and altitude. The values are the 20-year means (1987-2006) from the
GEOS-Chem simulation. The largest African influences appear in the Asian middle and upper
troposphere, i.e., 12-16 ppbv at 500 and 200 hPa. African ozone in the Asian lower troposphere is

reduced to 2-10 ppbv at 700 hPa. Unlike imported European and North American ozone that influences
Asia mostly over high altitudes, i.e., north of 30°N (Zhu et al., 2017b), imported African ozone prevails
over lower latitudes, generally between 5-40$^o$N, considerably larger than European and North American
ozone south of 30$^o$N (Wild et al., 2004; Sudo and Akimoto, 2007; Zhu et al., 2017b). North of 50$^o$N,
African ozone influence is small, i.e., less than 8 ppbv and 2 ppbv, respectively, above and below the

Asian middle troposphere. Seasonally, imported African ozone in the upper troposphere peaks in NH
spring around 30$^o$N (~16 ppbv, Fig. 5a) and is at its minimum in NH summer south of 25$^o$N. Owing to
the high radiative forcing efficiency, the change in ozone concentrations in the upper troposphere can
impact climate more significantly than that in the lower troposphere (Lacis et al., 1990). Therefore, the
influence of African ozone on the climate in southern Asia is likely larger than that of European and

North American ozone. In the middle troposphere (Fig. 5b), African ozone is at the maximum (~16
ppbv) in NH winter between 20 $^o$N and 25$^o$N. In the lower troposphere (Fig. 5c), between 5-40$^o$N,

African ozone is high in NH winter (~6-10 ppbv), while south of $10^{o}N$, African ozone has another peak in NH summer (~4 ppbv). Near the surface (Fig. 5d), African ozone concentrations are low, i.e. below 4 ppbv.

The strong seasonality of imported African ozone can also be shown vertically in Fig. 6a, in which imported African ozone is averaged over Asia south of $40^{o}N$. The fractional contribution of imported African ozone to ozone in Asia is shown in Fig. 6b. In the Asian upper troposphere, imported African ozone is largest during March-May (over 10 ppbv) and least during July-October (below 6 ppbv). In the Asia middle troposphere, imported African ozone is at a minimum from July to September (~4 ppbv, 6%

in the fractional contribution) and a maximum (over 10 ppbv, 14% in the fractional contribution) from January to March, about one month earlier than in the upper troposphere. In the Asian lower troposphere, imported African ozone is largest in NH winter (~4 ppbv, 8% in the fractional contribution) and lowest in NH autumn (~2 ppbv, 5% in the fractional contribution).

Furthermore, the total imported African ozone over Asia is divided by tropospheric layer (UT, MT,

and LT) and by hemisphere and the fractional contribution for each region is shown in Fig. 7. Over the Asian upper troposphere, African ozone from the NHAF UT accounts for 60-70% of the total imported African ozone (Fig. 7a), decreasing to ~20% in the Asian MT (Fig. 7b) and to below 20% in the Asian LT (Fig. 7c). In the meantime, the influence of African ozone from the NHAF MT becomes larger in the Asian MT and LT (20-40%). So does the influence of African ozone from the NHAF LT (20-40%).

African ozone from SHAF contributes to the total imported ozone throughout the year in the three tropospheric layers. The contribution is small, usually below 20% of the total imported African ozone, except for in NH winter over the Asian UT (Fig. 7a) and in NH summer over the Asian LT (Fig. 7c). The two exceptions in interhemispheric transport will be discussed in detail in section 4.2.

The seasonality of the transport of African ozone to Asia results from the collective impact of the

emissions of ozone precursors in the source region and the meteorology and chemistry from the source

region to the receptor region. The precursors of African ozone are mainly from biogenic, biomass burning, and lightning $NO_x$ sources (Piketh and Walton, 2004; Thompson, 2004; Aghedo et al., 2007; Giglio et al., 2013; Monks et al., 2015). The seasonalities of emissions from biogenic sources, biomass burning, lightning, and anthropogenic sources in Africa are characterized rather differently from the other continents (Williams et al., 2009; Guenther et al., 2012; Giglio et al., 2013; Albrecht et al., 2016). Since Africa covers areas in both hemispheres with a large portion in the tropics, the atmospheric circulation over Africa experiences obvious seasonal changes induced by the seasonality of the ITCZ and the Hadley cell (Nicholson, 2008, 2009; Žagar et al., 2011; Suzuki, 2011). To cast some light on the seasonal variations of imported African ozone over Asia presented in this section, the ITCZ and ozone precursor emissions over Africa are discussed in section 3.2. Based on the discussion, the possible mechanisms that modulate the transport of African ozone to Asia are speculated in section 3.3.

## 3.2 ITCZ and ozone precursor emissions over Africa

Based on Nicholson (2009, 2013), Suzuki (2011), and Žagar et al. (2011), the mean positions of the ITCZ in Africa in the four seasons are approximately illustrated in Figs. 8a-8d. The latitudinal migration of the ITCZ with season varies with longitude and the migration is within a wider range of latitudes in eastern Africa (10$^o$E-40$^o$E) than in western Africa (west of 10°E). In eastern Africa, ITCZ shifts between ~10$^o$S in NH winter and ~20$^o$N in NH summer, while in western Africa, the center of the ITCZ swings from 5$^o$N to 20$^o$N between NH winter and summer within the NH.

Seasonal variations of African ozone largely depend on biogenic, biomass burning, lightning and anthropogenic emissions. The anthropogenic emissions are generally considered to be small and have weak seasonality (Aghedo et al., 2007; Williams et al., 2009). Based on the emission inventories in GEOS-Chem that are described in section 2.1, the spatial distributions of ozone precursor emissions are shown by season in Fig. 8, including isoprene emissions from biogenic sources, CO emissions from

biomass burning, and $NO_x$ emissions from lightning at 700 and 300 hPa. Seasonal variations of these

emissions averaged over Africa, NHAF and SHAF are shown in Fig. 9.

     Isoprene ($C_5H_8$) is the dominant non-methane volatile organic compound (NMVOC) emitted by

vegetation (Marais et al., 2012). Biogenic isoprene emissions in Africa are considered to be responsible

for about 65% of ozone enhancement in the African upper troposphere (Aghedo et al., 2007). Isoprene

emissions shown in Figs.8a-8d are representative of biogenic emissions of ozone precursors. The

magnitude and spatio-temporal pattern of the isoprene emissions in Africa in Fig 9 are comparable with

Marais et al. (2014). The maximum biogenic isoprene emissions are over central African rainforests

throughout the year (Figs. 8a-8d). The seasonal cycle of biogenic isoprene over Africa peaks in NH

spring and autumn (Fig. 9a). Plenty of non-methane VOCs is emitted from biogenic sources that can be

uplifted by strong convection. Aghedo et al. (2007) and Zare et al. (2014) suggested that biogenic

emissions can lead to more ozone generation in the African upper troposphere than biomass burning and

anthropogenic emissions.

     In NH winter, fires in Africa are active in the NH between 0-10$^o$N and 15$^o$W to 40$^o$E (Figs. 8e-8h,

also see Sauvage et al., 2005). From NH winter to autumn, regions with biomass burning shift

southward (Figs. 8e-8h, also see van der Werf et al., 2010; Giglio et al., 2013). In NH summer, fires are

most active in the SH from the equator to 20$^o$S. Therefore, the regional CO emissions from biomass

burning peak during NH winter in NHAF and during NH summer in SHAF (Fig. 9b). Aghedo et al.

(2007) stated that biomass burning has the largest impact on surface ozone in the vicinity of the African

burning regions during the burning seasons.

In Africa, lightning $NO_x$ is produced mostly in the middle to upper troposphere (Figs. 8i-8p, also see

Pickering et al., 1998; Ott et al., 2010; Miyazaki et al., 2014). Miyazaki et al. (2014) estimated that the

altitudes where the annual lightning $NO_x$ emissions maximize are around 11 km in northern Africa and

9.36 km in southern Africa. Aghedo et al. (2007) suggested that lightning emissions mainly enhance

ozone production in the African middle and upper troposphere. Ascribe to the high efficiency of deep

moist convection, frequent lightning activities occur in the ITCZ (Christian et al., 2003; Avila et al., 2010). Collier and Hughes (2011) suggested that the peak lightning activities are generally located on the southern border of the ITCZ in Africa. The seasonality of the geographic variation of lightning $NO_x$ emissions clearly shows the influence of the ITCZ over Africa (Collier and Hughes, 2011). When the ITCZ reaches to its northernmost position in NH summer (Fig. 8c), lightning $NO_x$ emission over NHAF

becomes the highest (Figs. 9c and 9d). Similarly, the lightning $NO_x$ emission over SHAF peaks in NH winter (Figs. 9c and 9d).

### 3.3 Analysis of the mechanisms for the transport of African ozone to Asia

As African ozone mainly peaks in the Asian middle and upper troposphere, the horizontal distributions

of African ozone at 400 hPa in January, April, July, and October overlaid with winds are shown in Fig. 10 to illustrate the seasonality and the transport pathways of African ozone to Asia at this level. In NH winter, African ozone can be transported for a long distance along the subtropical westerlies in the two hemispheres, reaching the western Pacific in the NH and across Australia in the SH, respectively. In NH summer, the NH subtropical westerlies and tropical easterlies shift to their northernmost positions; less

African ozone can be transported to Asia than in the other seasons. Furthermore, Fig. 11 shows the latitude-altitude cross sections of African ozone and winds averaged from 0 to $40^o$E. This is to show how African ozone in the source region varies vertically along different latitudes. Fig. 12 is the same as Fig. 11 but for the longitude-altitude cross sections averaged from $20^o$N to $35^o$N so to show the transport pathway along longitude from Africa to Asia. Fig. 13 shows the 20-year mean paths for the

trajectories that run from seven reprehensive sites (Cairo, Ghat, Abuja, Khartoum, Juba, Dar es Salaam, and Luanda) to Asia. The mean paths are shown by season and by the original tropospheric layer, including the lapse day from the beginning of the trajectories. Additionally, we conduct three sensitivity

experiments by switching off the biogenic, lightning, and biomass burning emissions, respectively, to assist our analysis. The separate contributions of the three sources to tropospheric ozone over Africa are shown in Fig. S7. In the following, we analyze the information from these figures, in combination with literature, to explore possible mechanisms responsible for the transport of African ozone to Asia in the four seasons

### 3.3.1 In NH winter

In NH winter, the eastern part of ITCZ shifts to its southernmost position in eastern Africa around 15°S (Figs. 8a and 11a), while the western part of the ITCZ remains in NHAF around 5°N (Figs. 8a and 11a). In Fig 11a, the ITCZ around the two latitudes and the two cells of the Hadley circulation are clearly shown. Biomass burning is active in NHAF (Figs. 8e and 9b). Biogenic emissions (Fig. 9a) and $NO_x$ emissions from lightning (Fig. 9c) are the highest in SHAF. All these conditions are well reflected in Fig. 11a. The elevated African ozone in the NHAF lower troposphere from the equator to $10^o$N is resulted from high biomass burning and biogenic emissions (Figs. 8a and 8e, also see Figs. S7a and S7c, Aghedo et al., 2007), only this ozone is mostly confined under 700 hPa. In contrast, the high ozone concentrations over the SHAF middle and upper troposphere are due to deep convection and strong convergence of the ITCZ in SHAF that brings biogenic precursors to the upper troposphere and enhance ozone production there (Fig. 8a, also see S7a). In addition, ozone can be also generated in the middle and upper troposphere due to frequent lightning activities (Fig. 8i, also see Fig. S7b, Aghedo et al., 2007).

Driven by the Hadley cell, African ozone is transported upward over the ITCZ, northward in the middle and upper troposphere, and equatorward in the lower troposphere. The northward flow in the

middle and upper troposphere gradually weakens between 15-30$^o$N (Fig. 11a) where air parcels merge

into the NH westerlies. This is also seen in the HYSPLIT trajectories in Figs. 13a and 13b. The

trajectories originated from the two sites in the SH stop moving northward and turn toward the east

around 20$^o$N (Figs. 13a and 13b).

From the source region to the receptor region, the NH subtropical westerlies build the pathways (Fig.

12a, also see trajectories in Figs. 13a and 13b). In Fig. 12a, along the latitudinal pathways, downdrafts

behind the European trough around 40$^o$E divert part of African ozone to the surface. However, the

updrafts ahead of the European trough favor the uplift of African ozone so it can be transported for long

distance in the upper troposphere.

Finally, over the receptor region Asia, the downdrafts behind the Asian trough situated at around

140$^o$E bring African ozone from the upper layers to the lower layers. Consequently, the contribution of

African ozone to Asia becomes the highest in the middle and upper troposphere (Figs. 5 and 6). In NH

winter, the transport of African ozone to Asia generally takes 4-6 days varying with altitude and latitude

(Figs. 13a and 13b).

### 3.3.2 In NH spring

In NH spring, a region with high ozone concentrations above 40 ppbv appears in higher altitudes and

extends to a larger area in the middle and upper troposphere than in NH winter (Figs. 11b and 12b). This

region is further north than the elevated ozone region in NH winter. This is because the ITCZ in eastern

Africa shifts northward to near the equator (Fig. 8b). Biomass burning is least active in both NHAF and

SHAF (Figs. 8f and 9b) while the biogenic emissions in the season are the largest in NHAF and the

second largest in SHAF (Fig. 9a).As the ITCZ is over the region where biogenic emissions are also high

near the equator (Fig. 8b), The ITCZ effectively uplifts the biogenic precursors and also leads to

production of $NO_x$ from lightning in the upper layers (Fig.8j). The ozone precursors from both sources can enhance the generation of ozone in the middle and upper troposphere (Figs. S7d and S7e, also see Aghedo et al., 2007) where elevated ozone concentrations are apparent (Fig. 11b). Therefore, more African ozone can be exported out of Africa in the upper troposphere in NH spring than in NH winter (Fig.10a vs.10b, Fig.12a vs.12b). It takes more time for trajectories to arrive Asia in NH spring than in NH winter (Fig. 13a-13c vs. 13d-13f).

### 3.3.3 In NH summer

In NH summer, the ITCZ in Africa swings to its northernmost position around 15°N (Figs.8c and11c).The region with active biomass burning shifts to SHAF (Figs. 8g and 9b). A large amount of ozone, generated from the biomass burning in SHAF, is shown in Fig. 11c over the SHAF lower troposphere from ~15°S to the equator. However, this ozone is mostly confined in the lower troposphere (Figs. 11c and S7i). Note that ozone in the middle and upper troposphere north of 15°N is higher than in the other seasons, which is likely resulted from lightning activities and biogenic emissions (Figs. 8k, S2 and S7g). This ozone can be readily transported to Asia.

Along the transport pathway from Africa to Asia (Fig. 12c), Africa ozone concentrations at the source region are the highest among the four seasons (Figs. 10c and 12c). However, meteorological conditions along the pathway are most unfavorable for the transport of African ozone to Asia because of multiple reasons. First, the NH subtropical westerly jet in NH summer moves the northernmost position to around 40°N (Huang et al., 2012). The tropical easterlies also shift northward along (Fig. 8c) so to prevent African ozone from reaching Asia between 10°N and 30°N in the middle and upper troposphere (see Figs. 5 and 6 for imported African ozone concentrations, Figs. 10c and 12c for African ozone and wind fields). Second, on the transport pathway, the heavy downdrafts from the Saharan High, a

midtropospheric high-pressure system that is an eastward extension of the Azores High (Nicholson, 2017) and the Arabian High in the middle troposphere over Middle East (Liu et al., 2011) hamper African ozone from reaching Asia. Note in Fig. 12c, there is a region with lower African ozone than its surrounding in the lower troposphere between 10°E and 40°E. The downwards of African ozone near 30°E is likely due to a summertime trough at 40°E (Zhu et al., 2017b). Consequently, the amount of African ozone is reduced during the transport. Thirdly, in the source region, strong updrafts occur over the Tibetan Plateau (Fig. 12c) and further block the transport of African ozone toward the east. Finally, the strong divergence from the South Asia High obstructs the eastward transport of African ozone in the upper troposphere. For these reasons, imported African ozone over Asia is lowest in NH summer among the four seasons (Figs. 5 and 6). There are scarcely any trajectories from the African sites that can cross Asia, unlike in the other seasons (Figs. 13g, 13h, and 13i).

### 3.3.4 In NH autumn

In NH autumn, the ITCZ shifts southward to a location similar to in NH spring (Fig. 8b vs. 8d). Biogenic emissions are slightly lower than in NH spring (Fig. 8b vs. 8d, Fig. 9a), whereas lightning $NO_x$ emissions are higher than in NH spring (Fig. 8j vs. 8l, Fig. 9c). Biomass burning is strong but occurs mostly in SHAF (Fig. 9b) so that it imposes small influence on the Asian troposphere (Fig. 7). In NHAF, the strong biogenic emissions are uplifted effectively by the ITCZ, similar to NH spring. The uplifted biogenic precursors and $NO_x$ from lightning in the middle and upper troposphere lead to elevated ozone there (Fig. 11d, see also Figs. S7j and S7k). African ozone concentrations in the NHAF middle and upper troposphere look higher than in NH spring (Fig. 10b vs. 10d, Fig. 11b vs. 11d, Fig. 12b vs. 12d). However, there is less African ozone arriving Asia in NH autumn than in NH spring (Figs.

5 and 6). This may be due to two reasons. In NH spring, the elevated African ozone in NHAF is located in higher altitudes than in NH autumn (Fig. 11b vs. 11d, Fig. 12b vs. 12d). This ozone can be more effectively transported to Asia by more speedy winds in the upper layers. The second reason is because of the weaker subtropical westerlies in NH autumn than in NH spring (Fig. 10b vs. 10d, Fig. 12b vs.

12d, also see Huang et al., 2012). The transport pathways and the time for the transport of African ozone to Asia in NH autumn are similar to in NH spring, as shown in the trajectories (Fig. 13).

## 4 Meteorological influences on the interannual variation of the transport of African ozone to Asia

### 4.1 The influence of the ITCZ on African ozone transport to Asia in NH winter

As discussed, the ITCZ can impact meteorology in Africa (Sultan and Janicot, 2000; Xie, 2004; Hu et al., 2007; Collier and Hughes, 2011; Suzuki, 2011). The deep convection along the ITCZ can carry ozone precursors from biogenic, biomass burning, and anthropogenic emissions to the upper layers from the surface. The ITCZ is also a zone with large lightning activities and thus can impact the seasonality of $NO_x$ emission from lightning. The convective divergence in the upper troposphere over the ITCZ in

Africa plays a significant role in output of African ozone and in the interhemispheric transport between SHAF and NHAF (Fig. 11 and see trajectories in Fig. 13).

    To investigate the impact of ITCZ on the interannual variation of the transport of African ozone to Asia, we use the monthly Outgoing Longwave Radiation (OLR) data from NCAR at $2.5^o \times 2.5^o$ with temporal interpolation (Liebmann and Smith, 1996) as a proxy to indicate the intensity of the ITCZ.

According to Waliser et al. (1993) and Fukutomi and Yasunari (2013), the number of the grids with $OLR \leq 260$ W/m$^2$ in the region of $15^o$W-$45^o$E, $20^o$S-$20^o$N can indicate the intensity of the convection over the ITCZ in Africa.

    We find that intensity of ITCZ in Africa is mostly related to imported African ozone over Asia in NH winter. Fig. 14 shows the time series of anomalies of imported African ozone averaged over Asia,

against the intensity of the ITCZ over Africa. The intensity of the ITCZ is normalized, i.e., the normalized value is equal to the original value minus the mean and then divided by the standard deviation in January from 1987 to 2006. The intensity of the ITCZ and anomalies of imported African ozone are correlated with the correlation coefficient ($r$) of 0.61, 0.46, and 0.64 at the Asian lower (975 hPa), middle (600 hPa), and upper (200 hPa) troposphere, respectively, all statistically significant at the 95% level (p<0.05).

Separating by the hemisphere, significant correlations are also found between the ITCZ and imported NHAF ozone in the entire troposphere in Asia (Fig. 14). The interhemispheric transport of ozone from SHAF to Asia also correlates with the intensity of ITCZ in Africa well, with $r$ being 0.56 in the Asian upper troposphere (Fig. 14a). The interhemispheric transport to Asian middle and lower troposphere is little so their time series are not shown.

Overall, when the intensity of the African ITCZ is stronger, more ozone and ozone precursors are uplifted to the middle and upper troposphere and transported northward and then eastward to Asia by the NH subtropical westerlies. Additionally, driven by the enhanced convective divergence over the ITCZ, more ozone from SHAF is transported across the equator and to the NHAF upper troposphere. Consequently, African ozone increases in the Asian middle and upper troposphere. Meanwhile, carried by the downdrafts from the Asian winter monsoon (Zhu et al., 2017b), more ozone is transported to the surface in Asia.

**4.2 The influence of meteorology on the interhemispheric transport of ozone from SHAF to Asia**

As shown in Fig. 7 and discussed in earlier sections, ozone generated in SHAF can be transported across the equator and eventually to Asia. This is illustrated in more detail in Fig. 15, showing seasonal-altitude variations of imported ozone from NHAF and SHAF over Asia and the fractional contributions of NHAF and SHAF ozone to the total imported African ozone. Ozone from NHAF accounts for over

80% of the total imported African ozone in the Asian tropospheric column throughout the year, except for in the upper troposphere during NH winter and in the lower troposphere during NH summer (Figs. 15b and 15d). This represents two important interhemispheric transport pathways.

For the first transport pathway, Fig. 7 suggests that the SHAF ozone originates mainly from the SHAF UT. This ozone can be transported northward across the equator along the Hadley circulation and then eastward to Asia by the NH subtropical westerlies (Fig. 13a). The amount of ozone being transported is at the maximum in NH winter and at the minimum in NH summer (Figs. 7 and 15c) when the ITCZ is at its southernmost and northernmost position, respectively (Figs. 8a-8d). This can be further illustrated in the horizontal distribution of SHAF ozone at 200 hPa (Fig. 16a). SHAF ozone is 2-4 ppbv over China south of 30°N and 4-6 ppbv over western India. As shown in section 4.1 (Fig. 14a), the variation of the ITCZ intensity in Africa can explain 31% the interannual variation of the transport of SHAF ozone to the Asian in NH winter along this pathway.

The second transport pathway is shown in Fig. 16b for SHAF ozone distribution at 850 hPa in July, as Fig. 7c also suggests that the interhemispheric transport mainly occurs from the SHAF lower and middle troposphere. SHAF ozone concentrations are 2-4 ppbv over the Arabian Sea and the west coast of the Indian subcontinent. This ozone is transported to India in NH summer by the Somali cross-equatorial flow (Fig. 16b), which is the strongest seasonal cross-equatorial flow in the lower troposphere, serving as an essential component of the Asian monsoon system (Zhu, 2012). This is the reason for the maximum SHAF ozone (~4 ppbv) over the Asian lower troposphere south of 10°N in NH summer (Fig. 5c). One more evidence for this transport is shown in the trajectories from Dar es Salaam, eastern coast of Africa (Figs. 13h and 13i). The interhemispheric ozone transport to western India takes more than 6 days. Furthermore, the signal of the transport is captured in the ozonesonde data in western India. The vertical distributions of the seasonal ozone variations at Poona and Thiruvananthapuram are shown for the ozonsonde and GEOS-Chem data (Fig. 4). A dip of lower tropospheric ozone

concentrations in both data is apparent in NH summer, when the Somali jet carries clear air masses from sea which can be traced back to SHAF (Figs. 13h, 13i and 16b).

To search for a connection between the Somali jet and the imported SHAF ozone over western India, we use an index, proposed by Li et al. (2014a), to indicate the intensity of the Somali jet. The index is calculated as the mean meridional wind at 850 hPa in the domain shown in Fig. 16b. Li et al. (2014a) correlated the Somali jet and other cross-equatorial flows with the index. Fig. 17 shows the anomaly of SHAF ozone averaged in the lower troposphere at Poona and Thiruvananthapuram during NH summer

from 1987 to 2006. Positive correlations are found between the anomaly and the intensity of the Somali jet at both sites with the correlation coefficients over 0.56, significantly at the 95% level.

## 5 Discussion and conclusions

We have characterized the transport of African ozone to Asia according to the simulations of a global

chemical transport model, GEOS-Chem, for 20 years from 1987 to 2006. The ozone generated in the African troposphere is tracked using the tagged tracer simulation with GEOS-Chem. Combining with analysis of trajectory simulations using HYSPLIT and meteorological data of winds and OLR, we draw conclusions with discussion as follows.

1.    In Asia, imported African ozone shows strong seasonality that varies greatly with latitude,

longitude, and altitude in the troposphere (Figs. 5- 6). The influence of African ozone mainly prevails in Asia south of 40°N. From 5-40°N, imported African ozone is largest from March to May (10-16 ppbv) and lowest during July-October (2-6 ppbv) in the Asian upper troposphere (Fig. 5a). In the middle troposphere, imported African ozone is at a maximum from January to March (10-16 ppbv) and at a minimum from July to September (2-4 ppbv). Near the surface, the

African influence is small (below 6 ppbv). Overall, the influence of African ozone peaks in the Asian middle and upper troposphere between 20°N and 30°N in NH winter and spring. Over 5-

40°N, the mean fractional contribution of imported African ozone to the overall ozone concentrations in Asia is largest during NH winter in the middle troposphere (~18%) and lowest in NH summer throughout the tropospheric column (~6%).

2. Both the tagged ozone and the trajectory simulations show that the Hadley cell connects the subtropical westerlies to form a transport route from Africa to Asia (Figs. 10-13). This is a primary pathway that occurs throughout the year. It takes 4-6 days for African ozone to reach Asia depending on the season and the initial altitude and latitude of the airmass, i.e., faster in NH winter than in NH summer, faster in higher altitudes than in lower altitudes, and faster in higher latitudes than in lower latitudes (Fig. 13). The second transport pathway only appears in NH summer that runs from eastern Africa near the equator to the Indian low troposphere (Figs. 13h and 13i). It takes 6 or more days for African ozone to reach Asia along this pathway.

3. The seasonality of African ozone influence on Asia results from the collective effects of meteorology, chemistry, and ozone precursor emissions in the source and receptor regions and between them. For the primary transport pathway, ozone and ozone precursors from various sources in Africa can be efficiently lifted up to high altitudes by the ITCZ. The African ozone in the middle and upper troposphere can be transported northward along the upper branch of the Hadley circulation and then eastward to Asia along the NH subtropical westerlies in the middle and upper troposphere. Therefore, the seasonal swings of the Hadley cell and NH subtropical westerlies play a dominant role in determining the seasonality of this transport pathway. Consequently, imported African ozone in Asia is least in NH summer, increasing toward both NH spring and NH autumn. In NH spring, there are more biogenic and lightning $NO_x$ emissions than in NH winter. These precursors are uplifted by the ITCZ, making more ozone in the upper layers than in NH winter. Consequently, there is more African ozone to be transported to the Asian upper troposphere in NH spring than in winter (Figs. 5a and 6a). Although more ozone appears in Africa

in NH autumn than in spring, there is less imported African ozone over Asia in NH autumn than in spring (Figs. 5 and 6a), likely due to the facts that the elevated ozone in NHAF is located in higher altitudes in NH spring and the NH subtropical westerlies are stronger in NH spring. In NH summer, although the ozone outflow from Africa is high, the ozone hardly reaches Asia because

of the blockings of Saharan High, Arabian High, and Tibetan High along the transport pathway in the middle and upper troposphere, in addition to the northward swing of the westerlies. Finally, the ITCZ in Africa, combining with the geographic variations in ozone precursor emissions with season, can modulate the seasonality of transport of African ozone to Asia. When the ITCZ coincides with the ozone precursor emissions from biogenic and biomass burning emissions, in

addition to enhanced $NO_x$ emissions from lightning, strong ozone export out of Africa can be resulted, such as in NH spring.

4.  The interannual variation of the transport of African ozone to Asia is closely related to the intensity of the ITCZ in Africa in NH winter. Positive correlations are found between the intensity of the ITCZ in Africa and imported African ozone over Asia ($r = 0.64$ at 200 hPa, $r = 0.46$ at 600

hPa, and $r = 0.61$ at 975 hPa, and p<0.05 for the three layers) (Fig. 14). The stronger the ITCZ in a NH winter is, the more ozone and its precursors from the surface emissions can be uplifted. In the meantime, more lightning $NO_x$ is produced. Furthermore, the interhemispheric transport of ozone from SHAF is enhanced. Consequently, more African ozone can be transported to Asia.

5.  Ozone from NHAF makes up over 80% of the total imported African ozone in the Asian

troposphere in all layers and seasons, but with two exceptions in which ozone from SHAF becomes larger than 20% of the total imported African ozone (Figs. 7 and 15). The first exception occurs in the Asian upper troposphere during NH winter, corresponding to the primary transport pathway in NH winter (Fig. 7). In the season, the ITCZ swings to its southernmost position in Africa and the convective divergence over the ITCZ in the upper troposphere is enhanced,

resulting in more interhemispheric transport of ozone from SHAF. The interhemispheric transport

along this pathway is strongest in NH winter and weakest in NH summer (Figs. 7 and 15). The

second exception takes place in the Asian lower troposphere during NH summer. The SHAF

ozone is transported along the Somali jet, which is the second transport pathway, from eastern

Africa near the equator to India (Fig. 16b), forming an African ozone maximum in the lower

troposphere from the tropics to $15^{o}$N in NH summer (Fig. 5c). We find that the intensities of the

ITCZ in Africa and the Somali jet can respectively explain approximately ~30% of the

interannual variations in the transport of ozone from the southern hemispheric Africa to Asia in

the two cases (Figs. 14a and 17).

This study provides an enhanced understanding of the source-receptor relationship of ozone transport

from Africa to Asia. The findings on the transport routes from this study may also be applicable to other

atmospheric pollutants with similar lifetimes, such as carbon monoxide. Our analysis is based on the

simulations from the GEOS-Chem and HYSPLIT models, both of which have their own biases

associated with emission inventories, parameterization schemes, and input data. The influences of

African ozone can be further assessed by separating different emission sources.


*Acknowledgments.* We gratefully acknowledge that the GEOE-Chem model has been developed and

managed by the Atmospheric Chemistry Modeling Group at Harvard University. The HYSPLIT (Hybrid

Single-Particle Lagrangian Integrated Trajectory Model) model is developed by NOAA Air Resources

Laboratory, driven by the NCEP reanalysis data provided by NOAA/OAR/ESRL PSD, Boulder,

Colorado, USA. The ozonesounde data were acquired from the World Ozone and Ultraviolet Radiation

Data Center (http://www.woudc.org) under the World Meteorological Organization. The TES ozone

data are acquired from the NASA Langley Atmospheric Science Data Center. The NCEP/NCAR

reanalysis and OLR data are from NOAA Earth System Research Laboratory. This research is supported

by the Chinese Ministry of Science and Technology under the National Key Basic Research

Development Program (2014CB441203) and the Natural Science Foundation of China (41375140 and

91544230). We are indeed grateful to the anonymous reviewers for their valuable and helpful reviews.

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

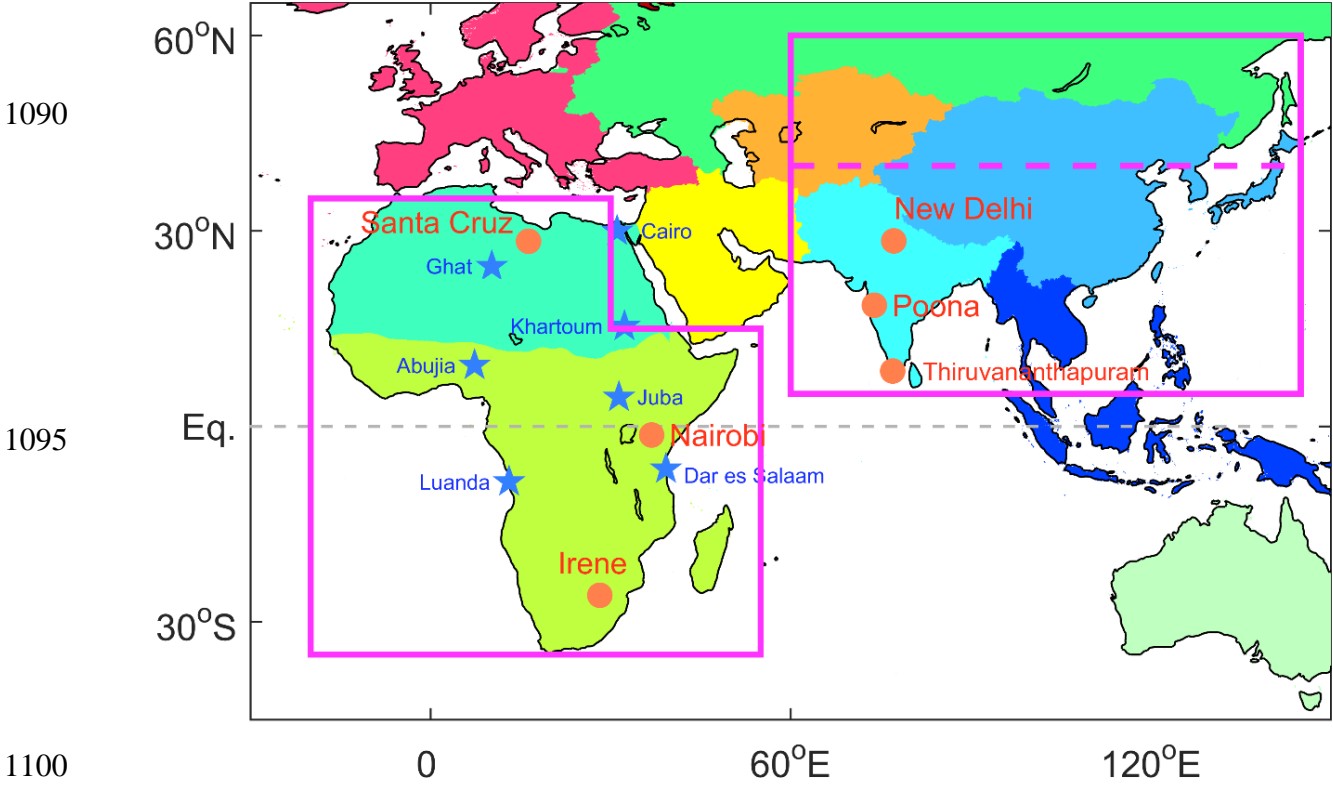

Figure 1. Domains of the source region, Africa, and the receptor region, Asia (areas within the solid purple lines). The area south of the pink dashed line at 40°N in Asia is used for calculating the regional mean of Asia for Figs. 6, 7, and 15. The brown dots indicate the locations of the ozonesonde sites where the ozone measurements are compared with the GEOS-Chem simulations. The seven blue stars indicate the sites where forward trajectory simulations are originated. The color-filled continents stand for the source/receptor regions defined in the HTAP Phase 2 (HTAP2).

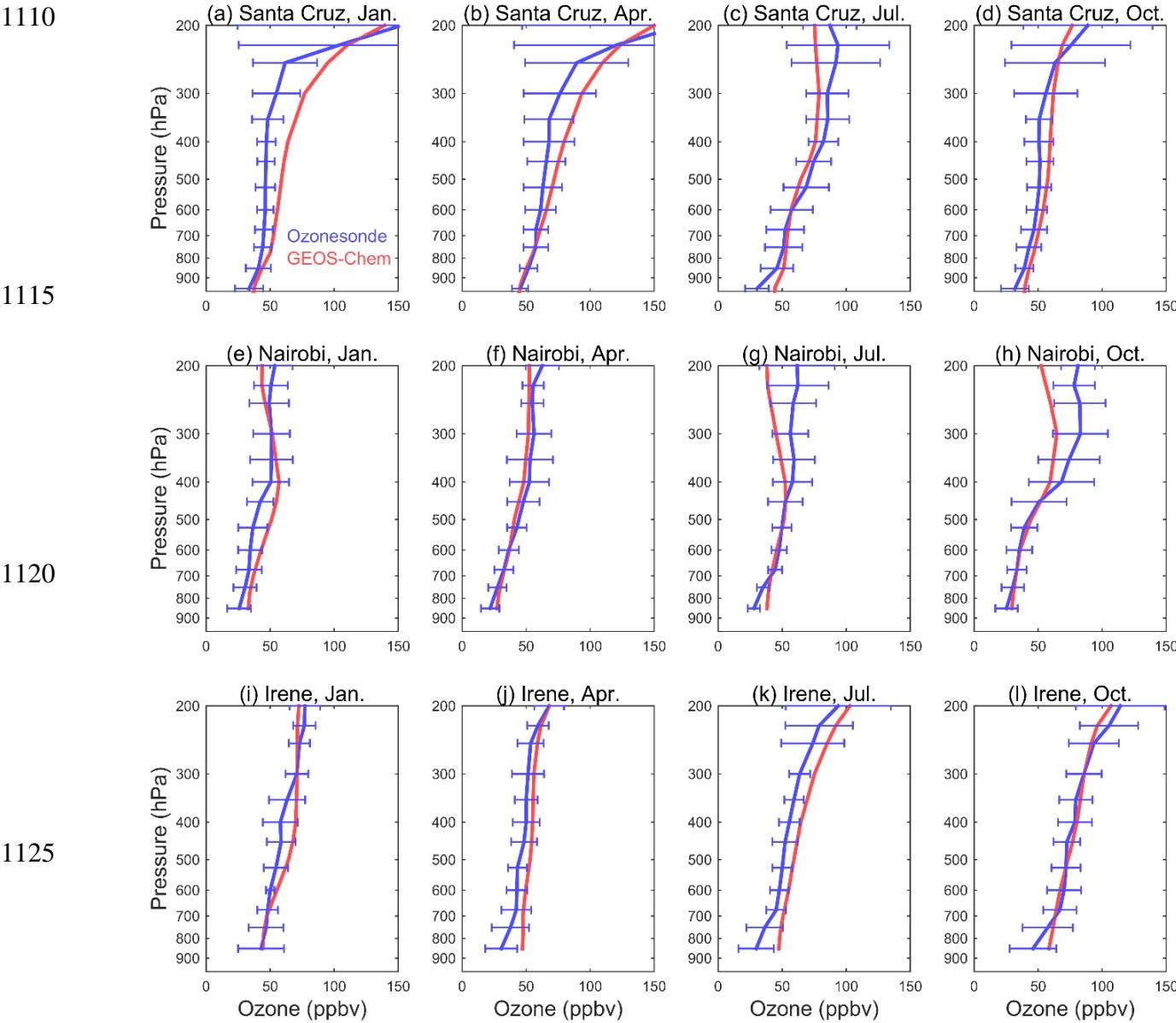

Figure 2. Comparison of the monthly mean ozone vertical profiles between the GEOS-Chem
simulations (red line) and the ozonesonde measurements (blue line) averaged over 1999-2003 at Santa
Cruz (1st row), over 2003-2006 at Nairobi (2nd row), and over 1999-2005 at Irene (3rd row), in January
(1st col.), April (2nd col.), July (3rd col.), and October (4th col.), respectively. The horizontal bar indicates
the standard deviation.

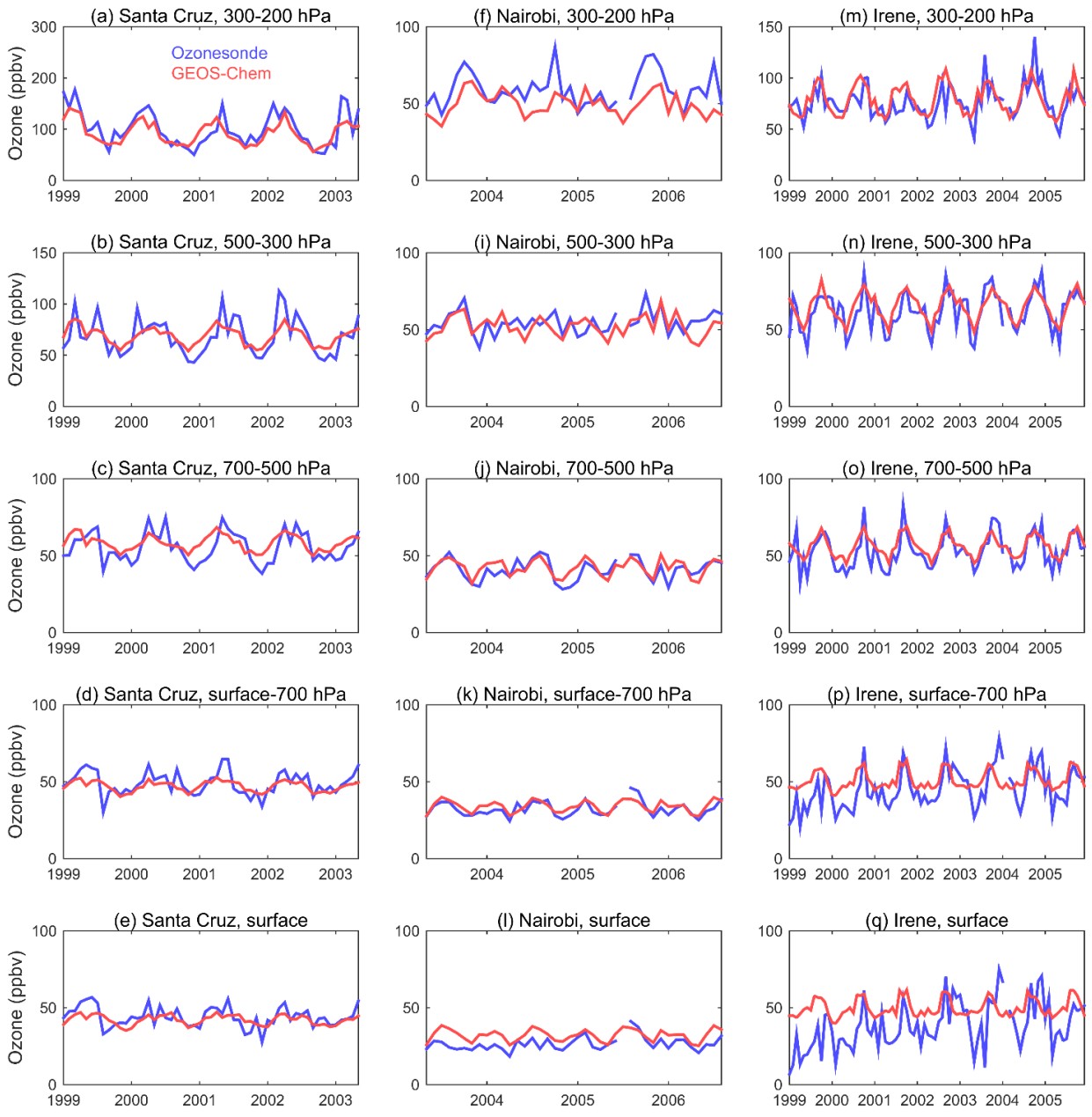

Figure 3. Comparison of the time series of monthly ozone between GEOS-Chem simulations (red line) and ozonesonde measurements (blue line) for layers averaged over 300-200 hPa (1$^{st}$ row), 500-300 hPa (2$^{nd}$ row), 700-500 hPa (3$^{rd}$ row), surface-700 hPa (4$^{th}$ row), and at the surface layer (5$^{th}$ row) at Santa Cruz (1$^{st}$ col.), Nairobi (2$^{nd}$ col.), and Irene (3$^{rd}$ col.) in Africa, respectively.

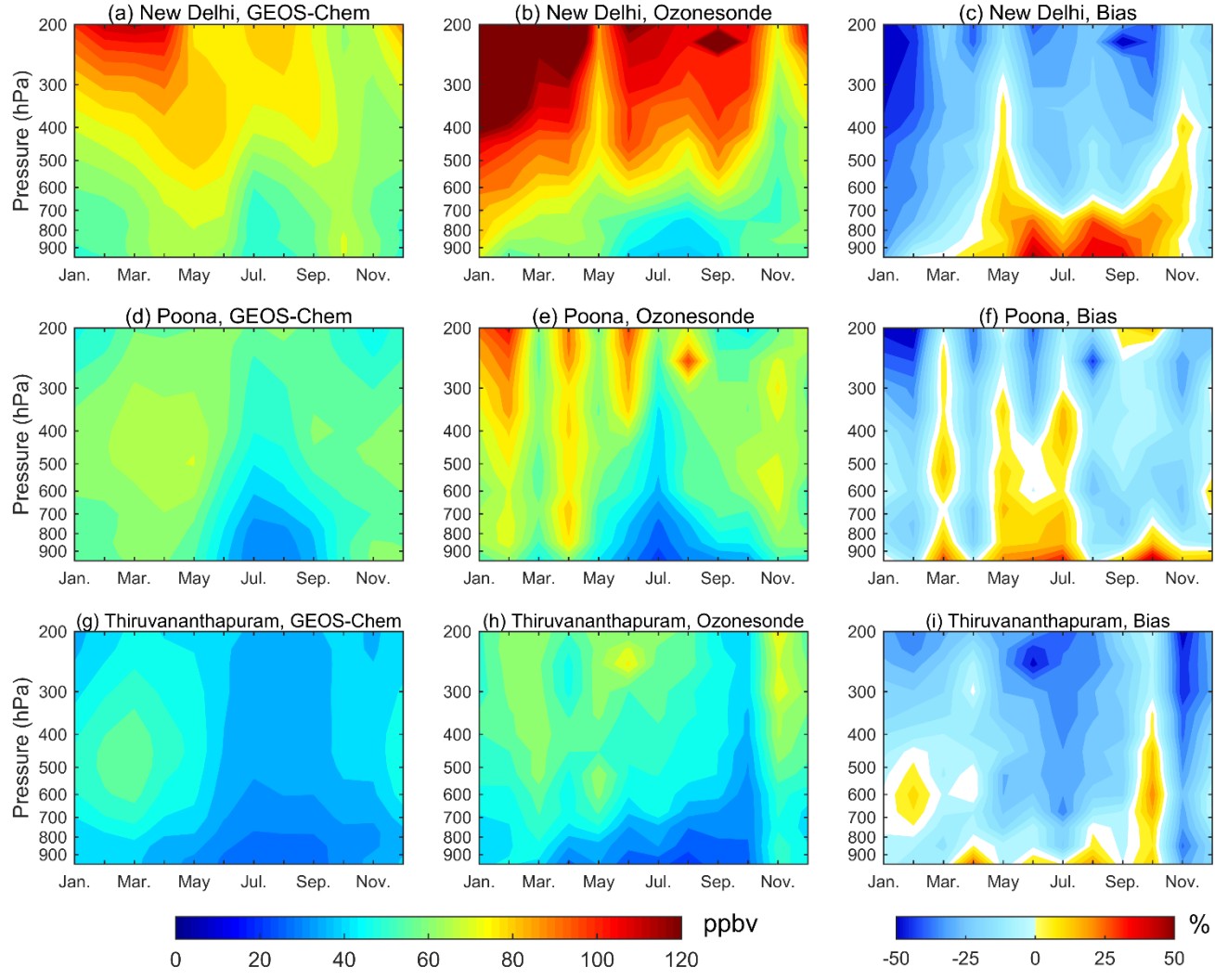

Figure 4. Seasonal variation of vertical ozone profiles from the GEOS-Chem simulations (in ppbv, 1st col), ozonesonde measurements (in ppbv, 2nd col.), and the corresponding bias (in %, 3rd col.) averaged over 1994-2003 at three Indian sites, New Delhi (1st row), Poona (2nd row), and Thiruvananthapuram (3rd row).

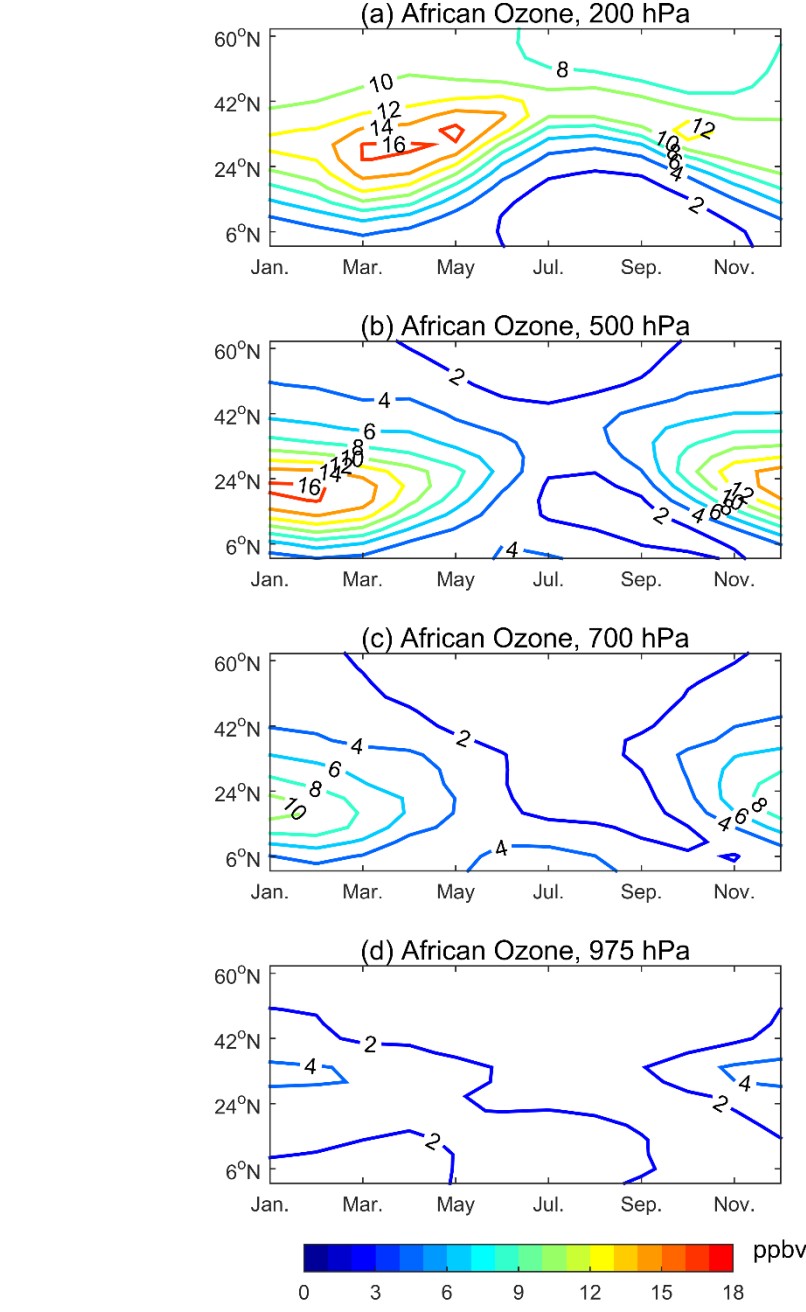

Figure 5. Seasonal variation of imported African ozone (in ppbv) over Asia varying with latitude at (a) 200 hPa, (b) 500 hPa, (c) 700 hPa, and (d) 975 hPa. The values are the 20-year means (1987-2006) from the GEOS-Chem simulation and averaged over 60-145°E.

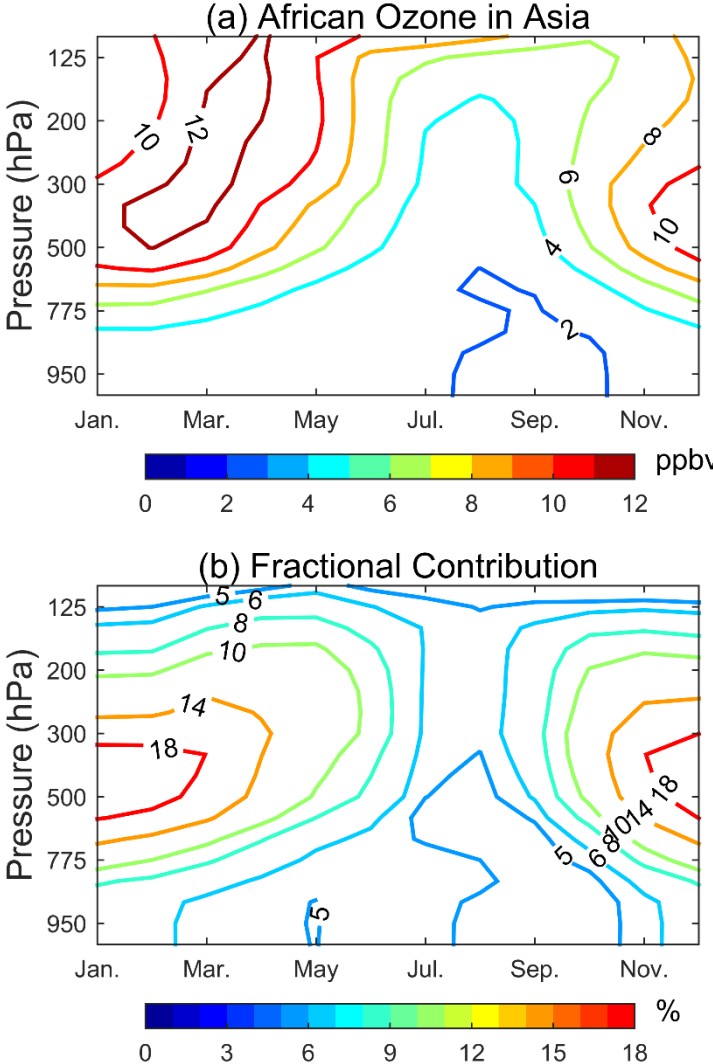

Figure 6. (a) Seasonal variation of imported African ozone (in ppbv) varying with altitude. (b) The same as (a) but for the corresponding fractional contribution (in %). The values are the 20-year means (1987-2006) from the GEOS-Chem simulation averaged over Asia (60-145°E, 5-40°N, see Fig. 1).

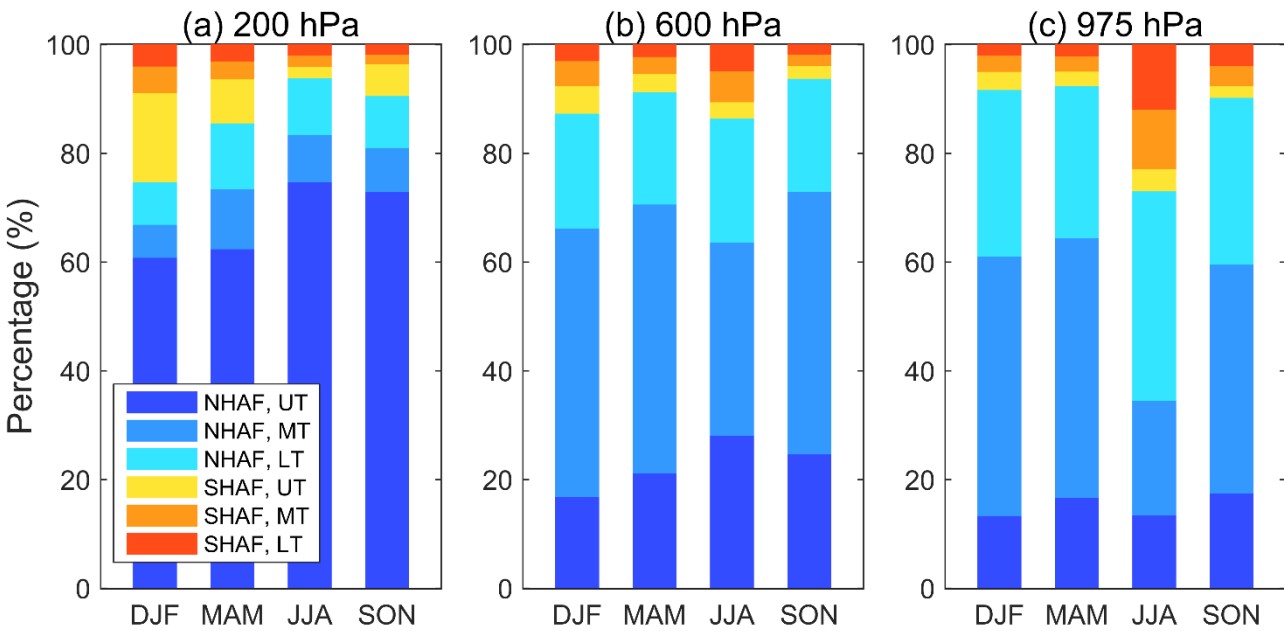

Figure 7. The fractional contribution of the imported African ozone from a layer in Africa to the total imported African ozone from all layers in Africa (in %). The layers include the lower troposphere (LT), middle troposphere (MT), and upper troposphere (UT). Each layer is further separated by hemisphere with blueish color for NHAF and reddish color for SHAF. The fractional contribution is shown in bars by season at (a) 200 hPa, (b) 600 hPa, and (c) 975 hPa over Asia. The values are the 20-year means (1987-2006) from the GEOS-Chem simulation averaged over Asia (60-145$^o$E, 5-40$^o$N, see Fig. 1).

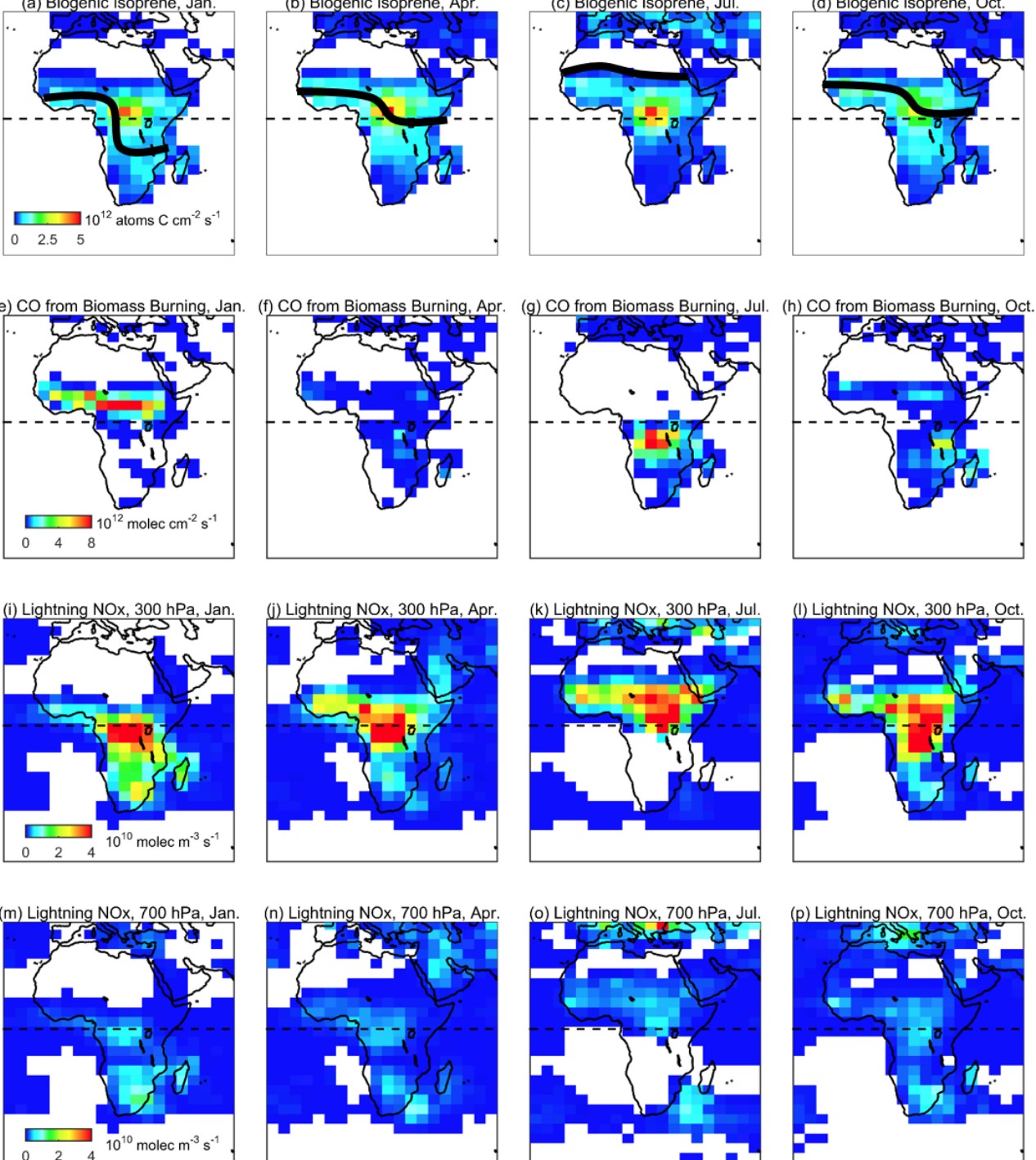

Figure 8. Distributions of isoprene emissions from biogenic sources (1st row), CO emissions from biomass burning (2nd row) and $NO_x$ emissions from lightning at 300 hPa (3rd row) and 700 hPa (4th row) by season in 2005. The data are based on the emission inventories in GEOS-Chem (see section 2.1). The dash lines indicate the equator. The black solid lines indicate the seasonal variation in the location of the ITCZ.

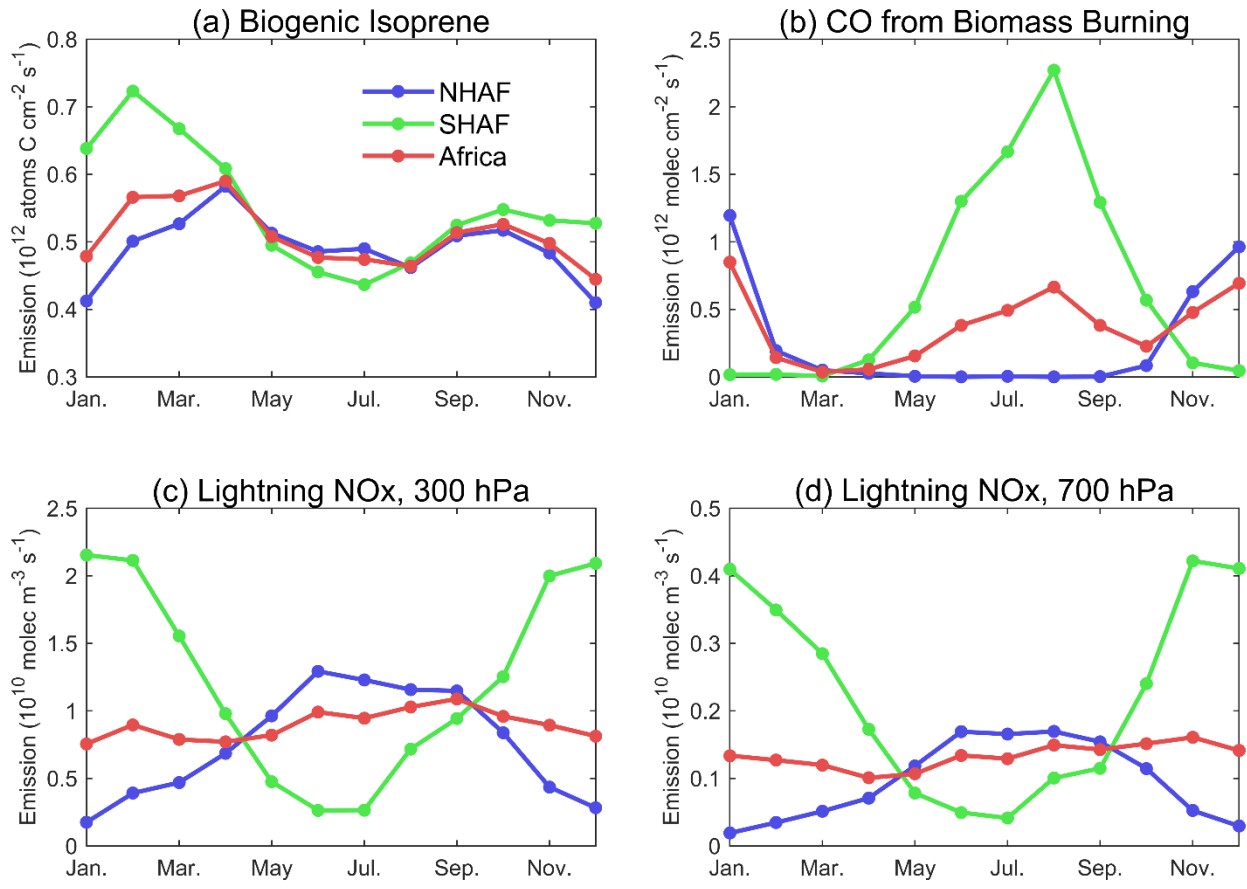

Figure 9. Seasonal variations for (a) isoprene emissions from biogenic sources, (b) CO emissions from biomass burning, NO$_x$ emissions from lightning at (c) 300 hPa, and (d) 700 hPa averaged over Africa, 1220 NHAF, and SHAF in 2005.

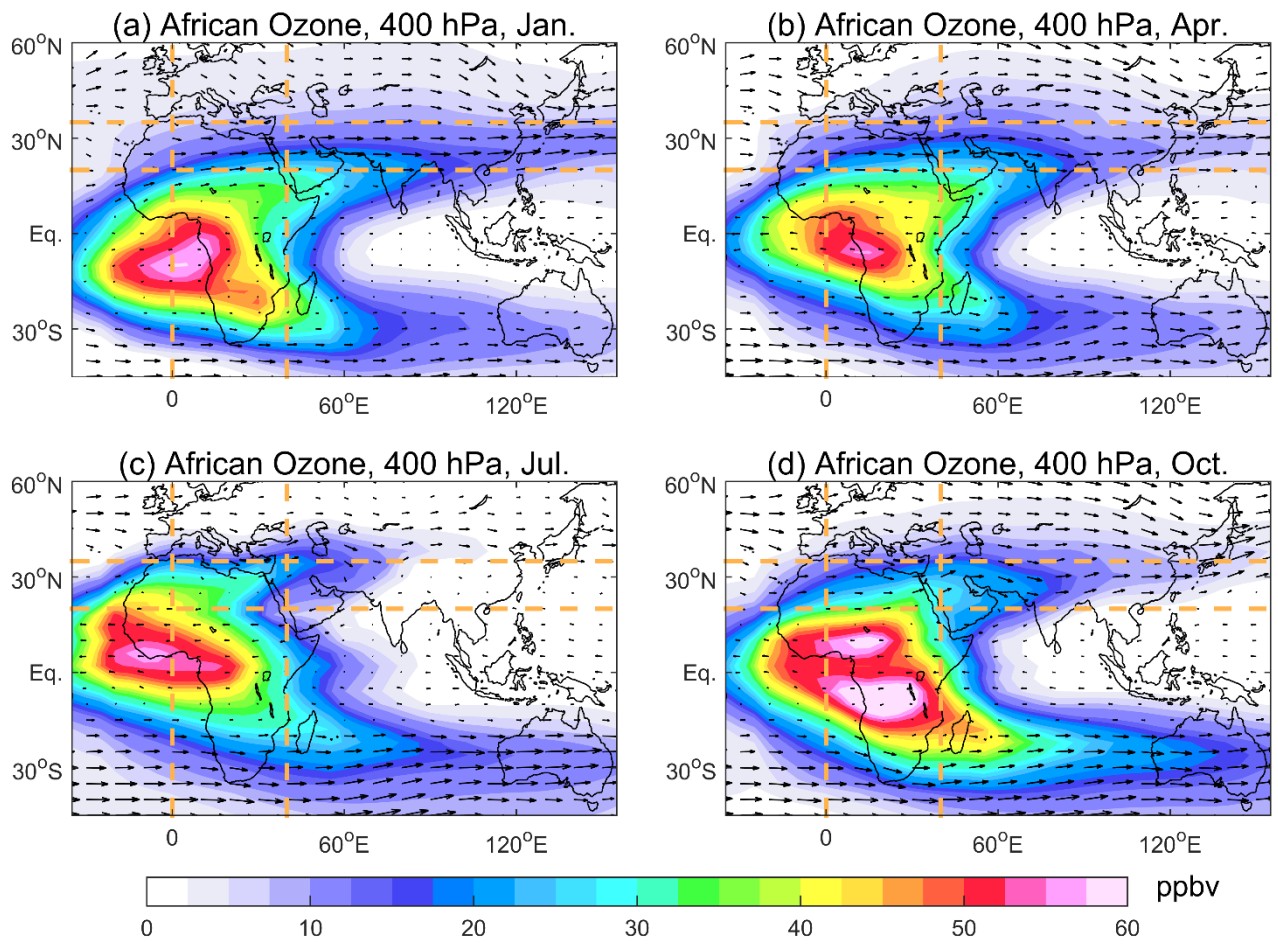

Figure 10. Horizontal distributions of African ozone (in ppbv, in color) overlaid with winds (in arrow) at 400 hPa in (a) January, (b) April, (c) July, and (d) October. The ozone values are the means from the 20-year GEOS-Chem simulation. Between the two pairs of parallel dished lines, latitudinal and longitudinal means are taken and shown in Fig. 11 and Fig. 12, respectively.

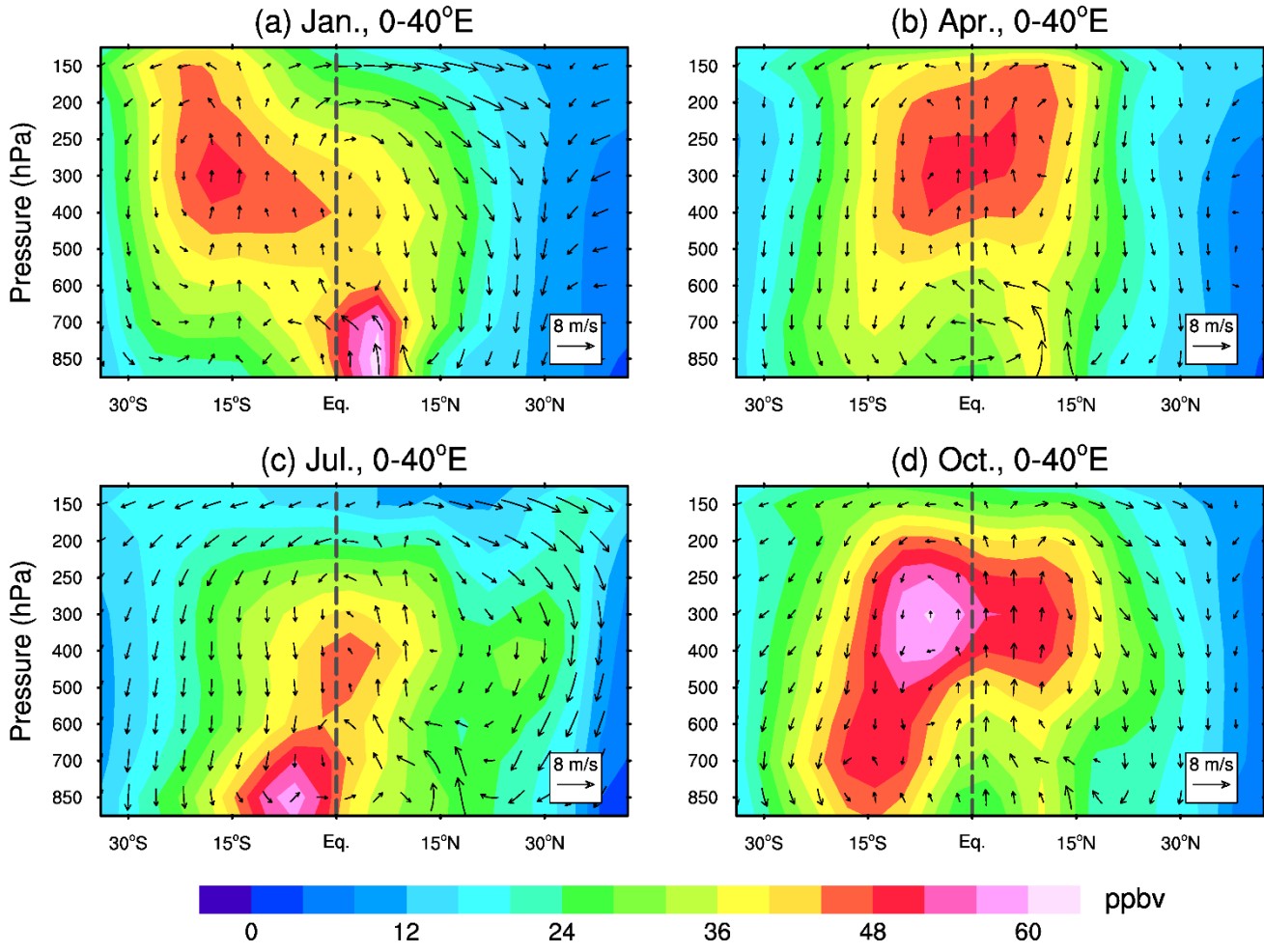

Figure 11.  Latitude-altitude distribution of African ozone (in ppbv, in color), overlaid with winds (in arrow) in (a) January, (b) April, (c) July, and (d) October. The ozone values are the means over 0-40°E (see Figure 10) from the 20-year GEOS-Chem simulation. The vertical velocities in the p-coordinates are enlarged by 200 times for illustration purposes. The vertical dashed line indicates the equator.

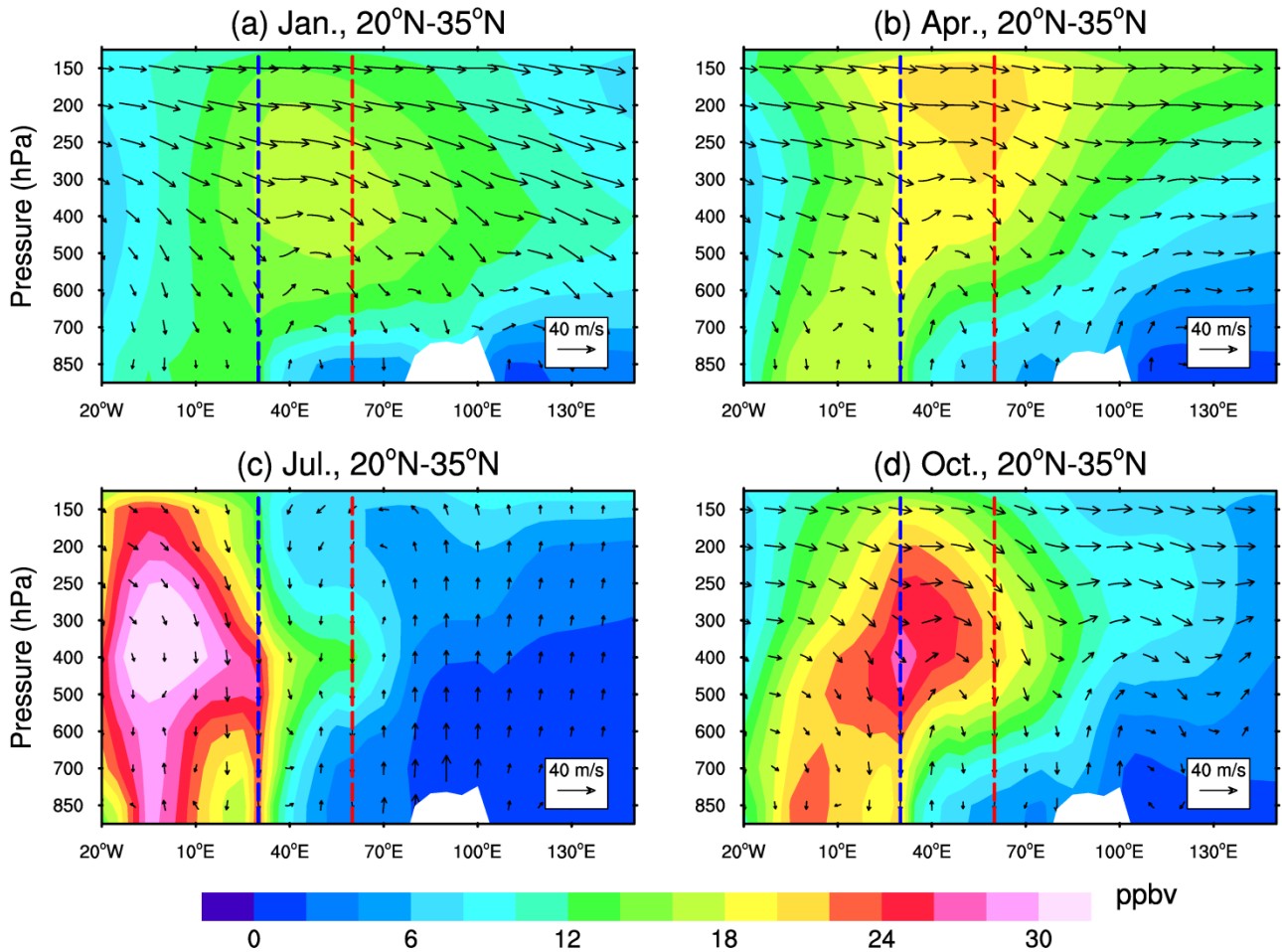

Figure 12. Longitude-altitude distribution of African ozone (in ppbv, in color), overlaid with winds (in arrow) in (a) January, (b) April, (c) July, and (d) October. The ozone values are the means over 20°N-35°N (see Figure 10) from the 20-year GEOS-Chem simulation. White areas indicate topography. The red and blue dash lines indicate the western border of Asia and eastern border of Africa, respectively. The vertical velocities in the p-coordinates are enlarged by 200 times for illustration purposes.

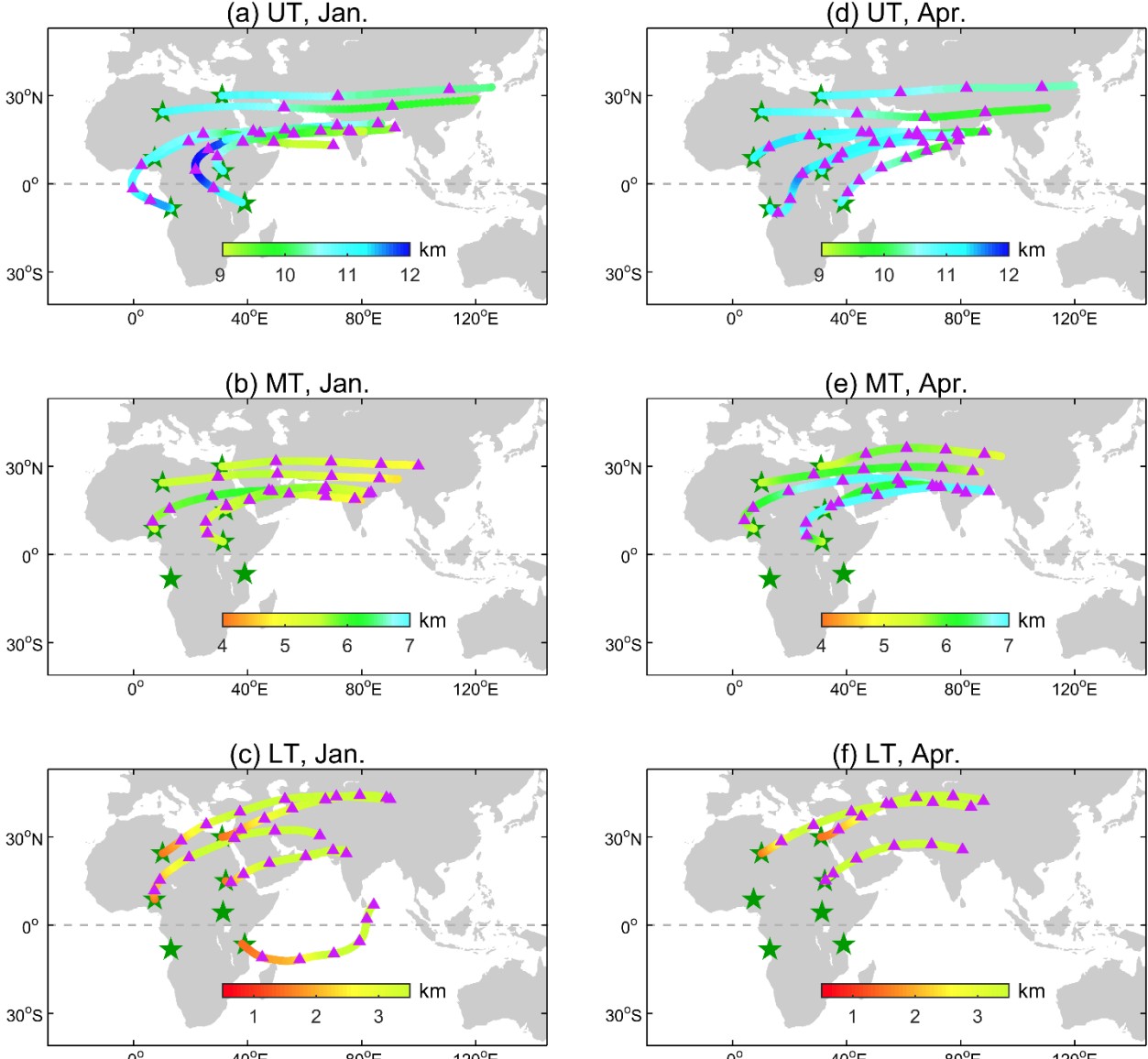

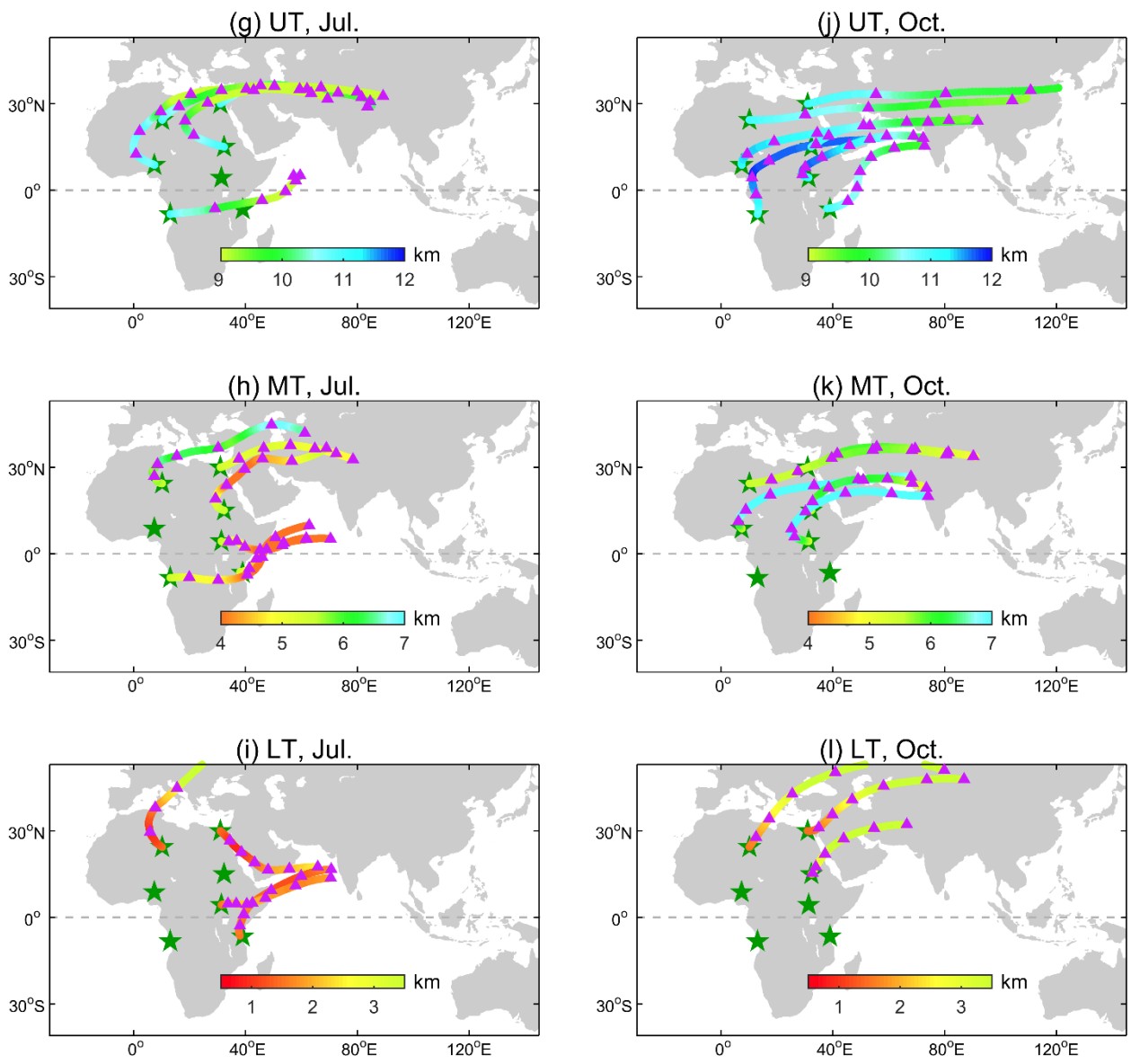

Figure 13. Mean seasonal paths between period 1987- 2006 for the trajectories that reach Asia and start from African sites at Cairo, Ghat, Abuja, Khartoum, Juba, Dar es Salaam, and Luanda in Africa (stars, also see Fig. 1) from 11 km (1st row), 5.5 km (2nd row), and 1.5 km (3rd row) in January (1st col.), April (2nd col.), July (3rd col.), and October (4th col.). The triangles along the trajectory paths indicate the lapse days from the beginning.

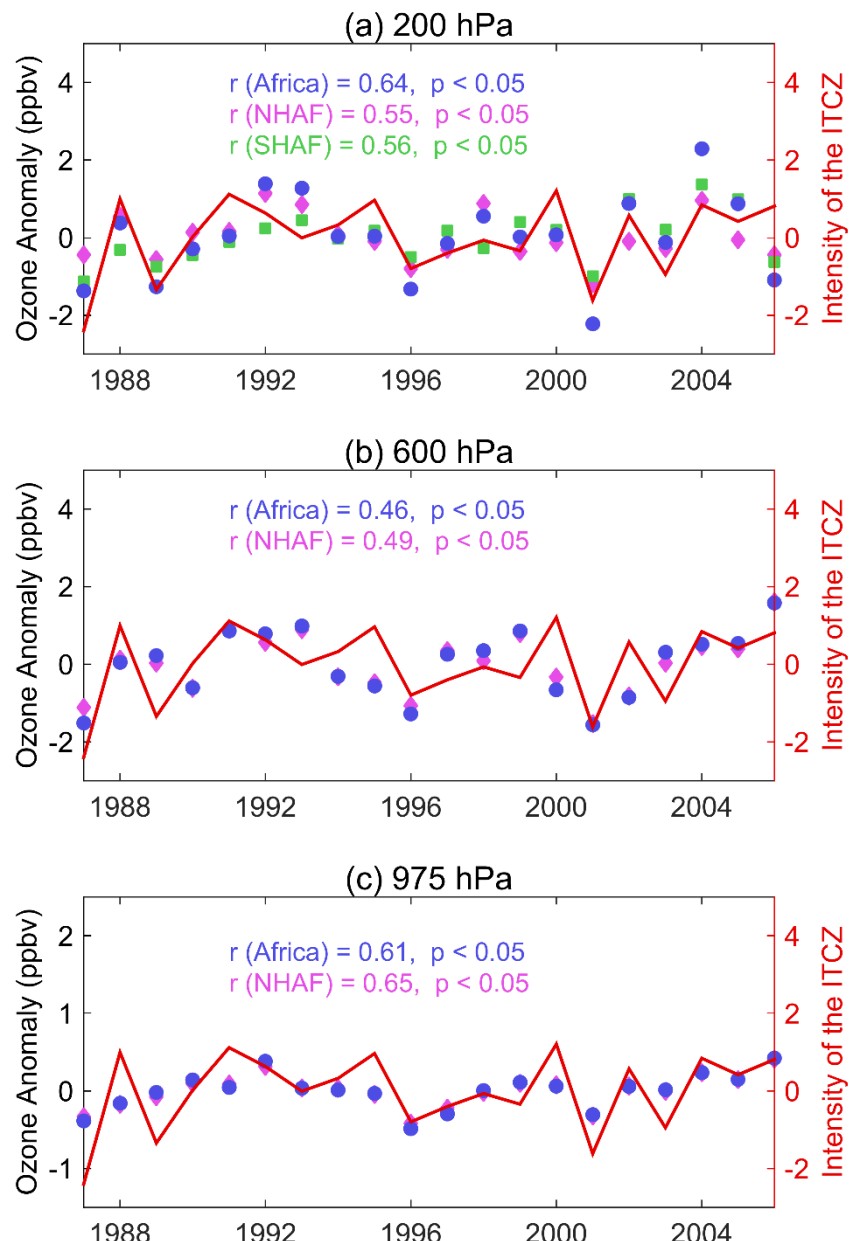

Figure 14. Interannual variation of the intensity of the ITCZ (the values are normalized) over Africa and the anomaly of imported ozone from Africa, NHAF and SHAF over Asia from 1987 to 2006 at (a) 200 hPa, (b) 600 hPa, and (c) 975 hPa in January, respectively.

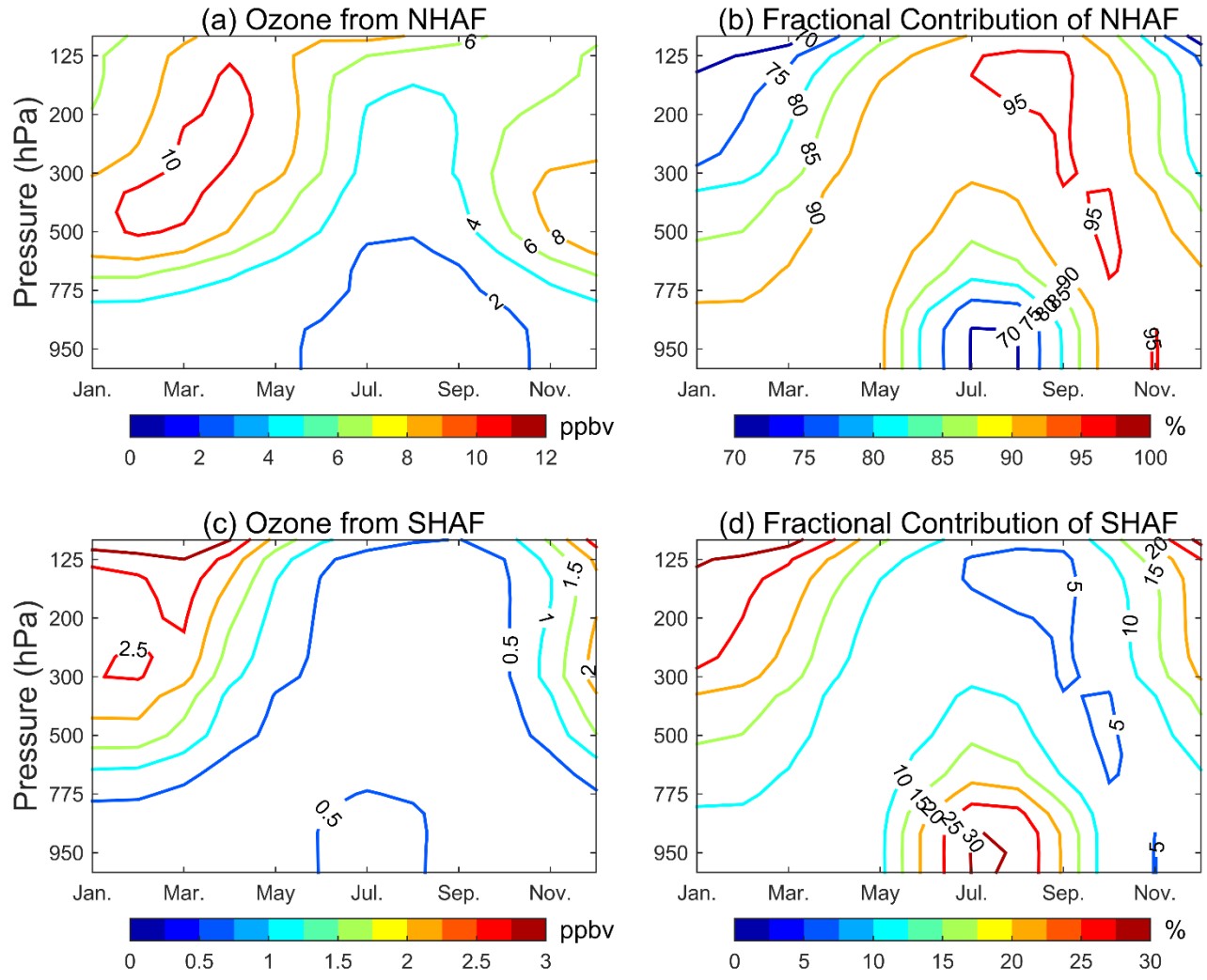

Figure 15. Seasonal-altitude variations of imported African ozone (in ppbv) from (a) NHAF and (c) SHAF over Asia (60-145$^o$E, 5-40$^o$N, see Fig. 1) and the fractional contribution of ozone from (b) NHAF and (d) SHAF to the total imported African ozone over Asia (in %). The ozone values are the means from the 20-year GEOS-Chem simulation.

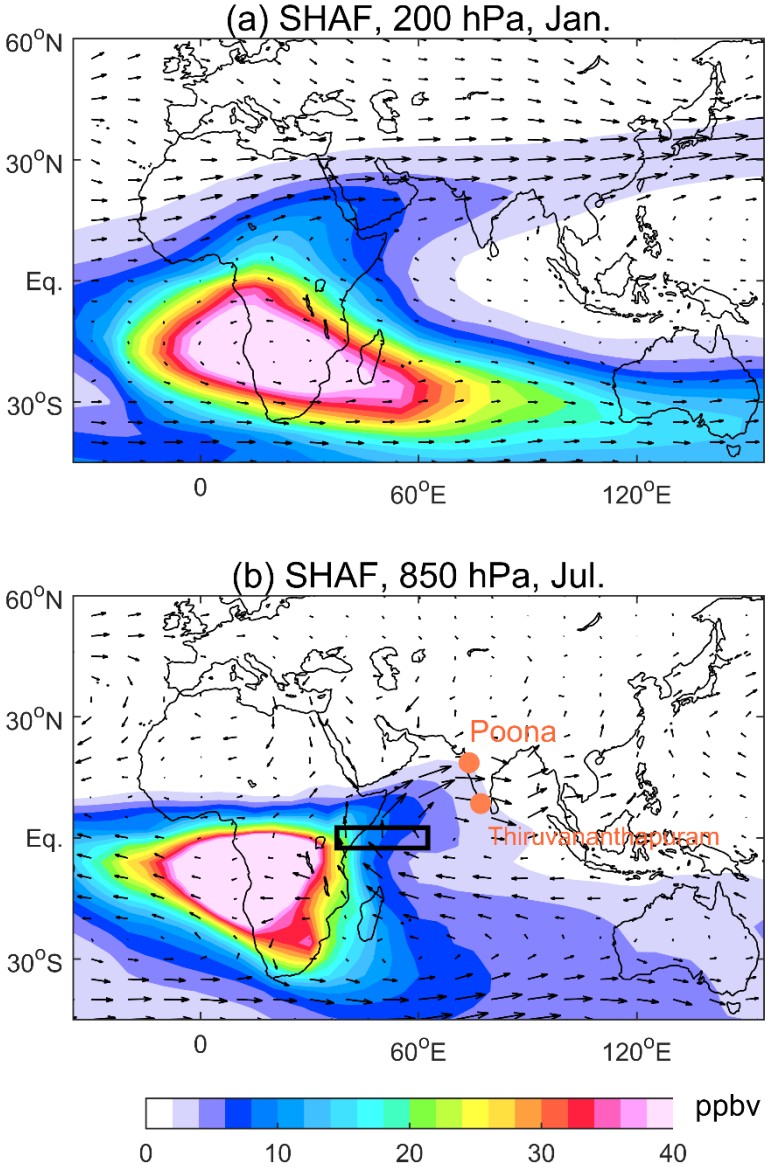

Figure 16. Distributions of ozone from SHAF (in ppbv, in color) overlaid with winds (in arrow) at (a)
200 hPa in January and (b) at 850 hPa in July. The ozone values are the means from the 20-year GEOS-
Chem simulation. The black rectangle indicates the domain used in calculating the strength of the
Somali cross-equatorial flow. The brown dots indicate the ozonesonde sites in western India.

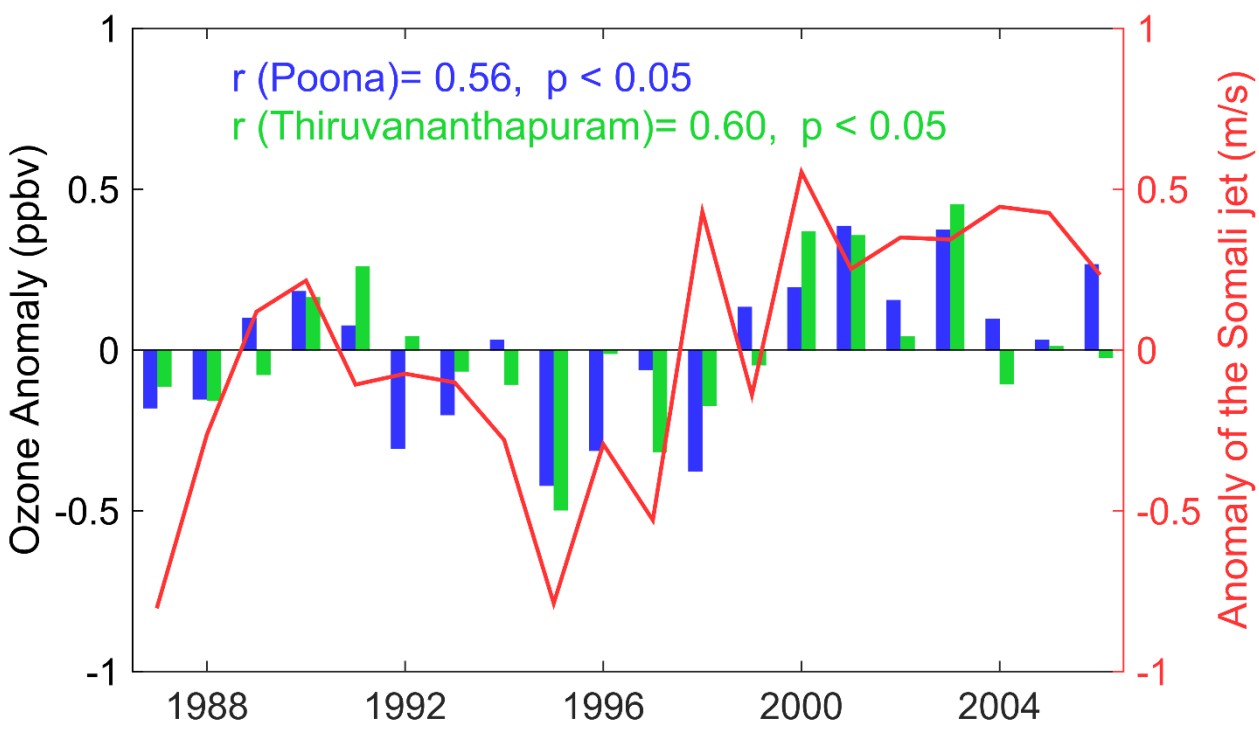

Figure 17. Interannual variation of the anomaly of ozone from SHAF at two Indian sites, Poona and Thiruvananthapuram in the lower troposphere in NH summer and the strength of the Somali jet from 1987 to 2006.

Table 1. The ozone vertical profiles (see Fig. 2) simulated with GEOS-Chem in comparison with the ozonesonde measurements at Santa Cruz, Nairobi, and Irene in Africa in terms of bias (in %), root-mean-square error (RMSE, in ppbv), and correlation coefficient (r) in the four seasons.

| | Month | Jan. | Apr. | Jul. | Oct. |
|---|---|---|---|---|---|
| Santa Cruz | r | 0.94 | 0.95 | 0.98 | 0.95 |
| | Bias (%) | 23.2 | 8.8 | -1.6 | 9.1 |
| | RMSE (ppbv) | 15.5 | 11.8 | 9.2 | 7.0 |
| | Month | Jan. | Apr. | Jul. | Oct. |
| Nairobi | r | 0.69 | 0.99 | 0.26 | 0.94 |
| | Bias (%) | 10.7 | -2.3 | -9.1 | -9.3 |
| | RMSE (ppbv) | 7.6 | 4.3 | 12.3 | 14.2 |
| | Month | Jan. | Apr. | Jul. | Oct. |
| Irene | r | 0.89 | 0.97 | 0.99 | 0.97 |
| | Bias (%) | 5.0 | 15.6 | 20.9 | 1.8 |
| | RMSE (ppbv) | 5.9 | 7.5 | 10.9 | 5.5 |

Table 2. The time series of ozone concentrations at different tropospheric layers (see Fig. 3) from the GEOS-Chem simulations and ozonesonde measurements at Santa Cruz, Nairobi, and Irene in Africa. The two datasets are compared in terms of bias (in %), root-mean-square error (RMSE, in ppbv), and correlation coefficient (r).

| | Layer | 300-200 hPa | 500-300 hPa | 700-500 hPa | Surface-700 hPa | Surface Layer |
|---|---|---|---|---|---|---|
| Santa Cruz | r | 0.79 | 0.75 | 0.57 | 0.62 | 0.60 |
| | Bias (%) | -4.2 | 7.1 | 7.3 | -2.3 | -2.7 |
| | RMSE (ppbv) | 22.9 | 12.7 | 8.4 | 6.3 | 5.6 |
| Nairobi | Layer | 300-200 hPa | 500-300 hPa | 700-500 hPa | Surface-700 hPa | Surface Layer |
| | r | 0.76 | 0.79 | 0.85 | 0.90 | 0.82 |
| | Bias (%) | -14.2 | -3.3 | 4.9 | 6.5 | 22.8 |
| | RMSE (ppbv) | 13.0 | 7.4 | 5.1 | 3.5 | 6.6 |
| Irene | Layer | 300-200 hPa | 500-300 hPa | 700-500 hPa | Surface-700 hPa | Surface Layer |
| | r | 0.70 | 0.82 | 0.82 | 0.61 | 0.39 |
| | Bias (%) | 5.7 | 7.3 | 8.6 | 20.7 | 60.3 |
| | RMSE (ppbv) | 13.6 | 8.6 | 7.8 | 12.3 | 18.7 |