# Peer review of "Characteristics of intercontinental transport of tropospheric ozone from Africa to Asia"

_Atmospheric Chemistry and Physics, 2017_

## Referee Comment (RC1) · Anonymous Referee #1 · 7 Nov 2017

General Comments:

The authors provide a detailed analysis for the long-range transport of tropospheric O3 from Africa to Asia. They indicated that African O3 have important influences on free tropospheric O3 over Asia, and the imported African O3 peaks in winter because of the shifts of transport and emission patterns. I recommend the paper for publication after consideration of the points below.

1) The paper isn't concise enough for me. For example, Section 5 provides a summary for the transport and emission processes, which is actually a repeat of Section 4.2. In addition, considering the small contribution from SHAF (shown by Figure 4), it may not be necessary to have an individual section (Section 4.3) to discuss its influence.

[Figure]

2) The discussion should be improved. The authors should explain why the seasonal variability of biogenic isoprene is so weak (Figure 6); and revise the discussion about the contributions from various sources (i.e. biogenic, biomass burning and lightning, Section 4.2).

Specific Comments:

1: Line 147-149 Are the O3 production and loss rates generated using the full-chemistry simulation with the current model settings or from other studies (Wang et al. 2998; Zhang et al. 2008)?

2: Line 149-153 It would be better to show these regions as boxes in the map (e.g. Figure 5). It is difficult to imagine the regions just based on these lat/lon numbers.

3: Line 155-157 Did the authors evaluate the possible influences from interannual variations of meteorology on chemistry?

4: Line 166-167 Is there any other station available? Why are these three stations selected?

5: Section 2.3 Is the meteorological data the same as used by the HYSPLIT model? If they are the same, it would be better to combine Section 2.2 with Section 2.3.

6: Line 237-239 The influence of African O3 to south America across Atlantic is discussed, but isn't shown in the Figure. It could be better to remove the discussion about the transatlantic transport here.

7: Line 262-269 Although may not be necessary to explain, I am just curious about the reason for the discrepancy between western and eastern Africa.

8: Line 276-277 Figure 6 shows significant seasonal variation for biomass burning CO. Surprisingly, the seasonal variation of biogenic isoprene is ignorable, which seems inconsistent with other study (e.g. Marais et al. 2014). Is it associated with the color scale?

On the other hand, the normalized magnitudes of seasonal variability (Figure 7) are comparable between CO and isoprene. Is it due to the usage of standard deviation in the calculation? The approach for normalization is confusing.

Marais, E. A., Jacob, D. J., Guenther, A., Chance, K., Kurosu, T. P., Murphy, J. G., Reeves, C. E., and Pye, H. O. T.: Improved model of isoprene emissions in Africa using Ozone Monitoring Instrument (OMI) satellite observations of formaldehyde: implications for oxidants and particulate matter, Atmos. Chem. Phys., 14, 7693-7703, https://doi.org/10.5194/acp-14-7693-2014, 2014.

9: Line 315-362 The discussion in this section is superficial. The authors discuss the contributions from various sources without detailed calculations. For example, the authors indicated: 1) "In boreal spring, a region with high ozone concentrations (>40 ppbv) appears in higher altitudes and . . . mainly due to the highest biogenic emissions in the NHAF" 2) "In boreal autumn, the locations of the ITCZ and the Hadley cell are similar to these in boreal spring. Ozone in the African middle troposphere ... attributed to stronger lightning NOx emission"

However, there is no evidence to demonstrate that the contributions from biogenic and lightning activities are evaluated carefully. The discussion is simply based on the spatial distribution of Figure 6. The biogenic and lightning activities are highly similar between spring and fall, and it is hard to explain why the spring-time O3 is biogenic dominant, whereas autumn-time O3 is lightning dominant.

10: Section 4.2 It seems that Figure 8 and Figure 9 are already sufficient for the discussion. I suggest to remove Figure 10 to make the paper more concise.

---

## Referee Comment (RC2) · Anonymous Referee #2 · 20 Nov 2017

This is a relatively straightforward analysis of the interplay of meteorological processes and atmospheric chemistry in venting out ozone and ozone precursors from Africa and reaching Africa. The manuscript is well written, the figures appropriate.

As the authors state- there is relatively little literature discussing Africa-to-Asia transport, so this is a welcome addition, despite it doesn't make use of the recommendation in the HTAP2 exercise to harmonize region definitions to allow comparability of results.

A minor remark is that I don't see terribly much added value of the trajectory analysis in figure 12.

Although some attempt has been made to demonstrate the model's ability to model ozone over Africa, I think this could be done more convincingly- there is meanwhile a

host of other observations (surface, aircraft, satellite tropospheric ozone columns) that could be explored. Are signals from African ozone visible in soundings over India?

The organization and discussion of methods could be somewhat more systematic.

I suggest that the authors explore somewhat further these aspects, and recommend the manuscript to be accepted after taking these major and minor comments below into account.

Minor comments.

l. 11-30 the abstract could be somewhat more explicit in describing the regions and attribution methodology.

l. 16 Replace boreal by NH winter. Or find better way of describing which months are discussed. Are the > and < really meant to express minima and maxima?

l. 30 I miss some statement on the relevance of this analysis. How much of the Asian ozone was produced in Africa or from African precursor emissions- where is it most important (not only vertical but also geographically.

l. 35 give reference time to which this RF estimate pertains.

l. 46 add: as well as a range of papers in the HTAP2 (Galmarini 2016) special issue.

l. 53 The issue is also very connected to legislative issues related to the control of ozone and ozone exceedance in the western states of the USA, e.g. as discussed in Huang et al. (2017; already cited).

l. 54 One reference on LRT transport between South Asia and East Chakraborty et al. Science of the Total Environment, 523, 2015

l. 73 . . . makes a contribution. . . how is contribution defined? Zero out of emissions? This is important because later you present a different method.

l. 117 which resolution is used for GEOS-CHEM; what was the underlying resolution of

the assimilation product. Importantly for this paper, how is convection parameterized, is there any evaluation over Africa of these process. Interhemispheric mixing and similar: refer to any relevant application of the model that demonstrates it is fit-for-purpose for this study. I realize that these are discussed later, but I would have expected these descriptions here.

l. 125- If I understand correctly the authors merge the EDGAR3.2 global inventory with regional ones. Which period? How do these inventories compare with e.g. the HTAP2 inventory for 2008/2010 in this special issue, or EDGAR4.2 products (for time series).

l. 135 What is the global lightning source strenght and specific for Africa. How does this compare to other studies.

l. 137 briefly describe what is the 'standard' tagged ozone method. Pro's and con's-limitations. Comes now later.

l. 138 I expected this description earlier.

l. 140- 143: better include with the GEOSCHEM description.

l. 150- what is the reason for not simply taking the 'african mask'- instead two blocks. I suggest adding a simple figure, showing these masks on top of a map (perhaps along with the HTAP2 definition of Africa).

l. 155 I think most papers that I know keep (anthropogenic) emissions constant- but that is not necessarily the same as keeping the production terms constant. What could be the impact?

l. 175- what about sfc observations, tropospheric residual from satellite. The comparison is fairly superficial

l. 201 Imported ozone=>try to describe more exactly what it is. Region-average abundance of imported ozone (imported ozone could be also the flux through the western border, for instance).

p. 295 /figure 6: I guess if the units are molec/cm2/s this pertains to the integrated amount over a model layer; otherwise it should rather be per cm3?

---

## Author Comment (AC1) · 27 Jan 2018

*We thank the reviewers for their constructive comments, suggestions, and corrections. In our revision, we have addressed the reviewer's concerns. The following is a one-to- one response to their questions.*

*Response to Comments by Anonymous Referee #1*

General Comments:

The authors provide a detailed analysis for the long-range transport of tropospheric $O_3$ from Africa to Asia. They indicated that African $O_3$ have important influences on free tropospheric $O_3$ over Asia, and the imported African $O_3$ peaks in winter because of the shifts of transport and emission patterns. I recommend the paper for publication after consideration of the points below.

1) The paper isn't concise enough for me. For example, Section 5 provides a summary for the transport and emission processes, which is actually a repeat of Section 4.2. In addition, considering the small contribution from SHAF (shown by Figure 4), it may not be necessary to have an individual section (Section 4.3) to discuss its influence.

*Thanks for the comments and suggestions. We have constructed the paper to make it more concise. The presentation is polished throughout the paper. Section 5 in the last version has been removed and section 4.2 in the last version (now section 3.2) has been polished. We have recognized the section on the interhemispheric transport of ozone from SNAF and added more analyses in this section. Therefore, we think it is better to keep this section.*

2) The discussion should be improved. The authors should explain why the seasonal variability of biogenic isoprene is so weak (Figure 6); and revise the discussion about the contributions from various sources (i.e. biogenic, biomass burning and lightning, Section 4.2).

*Thanks for the points. In the last version, the color scale for the seasonal variation of biogenic isoprene used in Figure 6 was not appropriate so that the seasonal variation was not shown apparently. We have edited the color scale so to better show the magnitude of the seasonal variation of biogenic isoprene (now Figure 7). The seasonal variation of biogenic isoprene is also shown in Figure 8 for the regional*

*means. The discussion about the contributions from various sources has been revised in section 3.3.*

Specific Comments:

1: Line 147-149 Are the $O_3$ production and loss rates generated using the full-chemistry simulation with the current model settings or from other studies (Wang et al. 1998; Zhang et al. 2008)?

*In this study, we generated the ozone production and loss rates from the full-chemistry simulation using the current model settings of GEOS-Chem v9-02. We have clarified this in this revision.*

2: Line 149-153 It would be better to show these regions as boxes in the map (e.g. Figure 5). It is difficult to imagine the regions just based on these lat/lon numbers.

*Thanks for your suggestion. We have added Fig. 1 to show the definition of the regions. The sites used in the GEOS-Chem validation and trajectory analysis are also shown in Fig. 1.*

3: Line 155-157 Did the authors evaluate the possible influences from interannual variations of meteorology on chemistry?

*Previous studies have shown that the interannual variation of meteorology has impact on the production and loss of ozone (using ENSO as an example: Sekiya and Sudo, 2012, 2014; Hou et al., 2016). Using 39-year simulations from the global chemical transport CHASER of two experiments by fixing the emissions of ozone precursors and ozone production/loss rate, Sekiya and Sudo (2012) shows that the influence from the interannual variations of meteorology on the impact of transport on tropospheric column ozone is greater than that of chemistry over most of the globe. Sekiya and Sudo (2014) further suggested that the impact of chemistry is comparable to the impact of transport in the tropics. Hou et al. (2016) shows that El Nino and La Nina show opposite effect on ozone production in the tropical Pacific.*

*In this study, we focus on the impact of the interannual variation of meteorology on*

*the transport of African ozone to Asia. Therefore, we keep the ozone production and loss rate fixed in one year and allow the meteorology to vary from year to year. To test our results, we have generated ozone production rate and lose frequency in other two separate years. Using these two sets of daily ozone production and loss frequency, two additional time series of 20-year simulations can be generated. The three time series show consistent interannual variation in transport of African ozone to the Asian troposphere, which is driven only by meteorology. The differences in the daily ozone production rate and lose frequency among the three data sets reflect partially the meteorological influence on ozone chemistry. This sensitivity test suggests that although meteorology also impacts ozone chemistry, our results on the interannual variation of African ozone transport that is simulated with fixed chemistry in a year appear robust.*

*Reference: Hou, X., B. Zhu, D. Fei, X. Zhu, H. Kang, and D. Wang (2016), Simulation of tropical tropospheric ozone variation from 1982 to 2010: The meteorological impact of two types of ENSO event, J. Geophys. Res. Atmos., 121, 9220–9236, doi:10.1002/2016JD024945.*

4: Line 166-167 Is there any other station available? Why are these three stations selected?

*GEOS-Chem simulations have been validated with ozonesonde data extensively, for example, in North America (Zhang et al., 2008; Zhu et al., 2017b), Europe (Kim et al., 2015), and East Asia (Wang et al., 2011; Zhu et al., 2017a; Zhu et al., 2017b). However, few studies have validated the simulations in Africa. We specifically validate the performance of GEOS-Chem over Africa for an enhanced confidence on our analysis. These three stations are selected for their representative locations and relative long records in Africa. In this version, the ozone data from three ozonesonde stations in India are added for the GEOS-Chem validation. In addition, the Tropospheric Emission Spectrometer (TES) satellite observations are compared with the GEOS-Chem simulation in the middle troposphere (see the supplement file).*

5: Section 2.3 Is the meteorological data the same as used by the HYSPLIT model? If they are the same, it would be better to combine Section 2.2 with Section 2.3.

*The meteorological data are the same as used by the HYSPLIT model. The two parts have been combined into Section 2.3.*

6: Line 237-239 The influence of African $O_3$ to south America across Atlantic is discussed, but isn't shown in the Figure. It could be better to remove the discussion about the transatlantic transport here.

*The discussion about the transatlantic transport here has been removed.*

7: Line 262-269 Although may not be necessary to explain, I am just curious about the reason for the discrepancy between western and eastern Africa.

*In general, the latitudinal position of ITCZ follows the sun. In eastern Africa, the seasonal migration of ITCZ with latitude is more symmetrical around the equator, while in western Africa, the migration is limited (Collier and Hughes, 2011). The seasonal migration of the ITCZ in western Africa is complicated. Generally, in NH summer, the convergence zone is formed by the flows from the Atlantic cold tongue and the Saharan heat low, locating around $20^oN$ (Nicholson, 2009, 2013). In NH winter, the anticyclonic wind from northern Africa converges with the southerly wind from Atlantic. The ITCZ over western Africa still stays in the continent (Nicholson, 2013). Therefore, the seasonal migration of the ITCZ in western Africa is within a narrower range of latitudes than in eastern Africa.*
*Reference: Sharon E. Nicholson, "The West African Sahel: A Review of Recent Studies on the Rainfall Regime and Its Interannual Variability," ISRN Meteorology, vol. 2013, Article ID 453521, 32 pages, 2013. doi:10.1155/2013/453521*

8: Line 276-277 Figure 6 shows significant seasonal variation for biomass burning CO. Surprisingly, the seasonal variation of biogenic isoprene is ignorable, which seems inconsistent with other study (e.g. Marais et al. 2014). Is it associated with the color scale? On the other hand, the normalized magnitudes of seasonal variability

(Figure 7) are comparable between CO and isoprene. Is it due to the usage of standard deviation in the calculation? The approach for normalization is confusing.

Marais, E. A., Jacob, D. J., Guenther, A., Chance, K., Kurosu, T. P., Murphy, J. G., Reeves, C. E., and Pye, H. O. T.: Improved model of isoprene emissions in Africa using Ozone Monitoring Instrument (OMI) satellite observations of formaldehyde: implications for oxidants and particulate matter, Atmos. Chem. Phys., 14, 7693-7703, https://doi.org/10.5194/acp-14-7693-2014, 2014.

*Thanks for the points. Yes, the small seasonal variation in the old Figure 6 for the seasonal variation of biogenic isoprene is indeed due to the use of the color scale. We have edited the color scale in Fig. 7so that the seasonal variation of biogenic isoprene in Africa is better presented. The biogenic emission peaks in spring and autumn. The magnitude of biogenic isoprene is comparable to the results in Marais et al. (2014). Fig. 8 shows the seasonal variability of isoprene and CO. The units are the same as them in Fig. 7. Normalization is not taken any more in this revision to avoid confusing.*

9: Line 315-362 The discussion in this section is superficial. The authors discuss the contributions from various sources without detailed calculations. For example, the authors indicated: 1) "In boreal spring, a region with high ozone concentrations (>40 ppbv) appears in higher altitudes and … mainly due to the highest biogenic emissions in the NHAF" 2) "In boreal autumn, the locations of the ITCZ and the Hadley cell are similar to these in boreal spring. Ozone in the African middle troposphere ... attributed to stronger lightning NOx emission" However, there is no evidence to demonstrate that the contributions from biogenic and lightning activities are evaluated carefully. The discussion is simply based on the spatial distribution of Figure 6. The biogenic and lightning activities are highly similar between spring and fall, and it is hard to explain why the spring-time $O_3$ is biogenic dominant, whereas autumn-time $O_3$ is lightning dominant.

*Thanks for the comments. Aghedo et al. (2007) has shown that the biogenic and lightning emissions are the two important sources influencing African middle and*

*upper tropospheric ozone and affecting global tropospheric ozone burden. To further explore the differences between the situations in NH spring and in NH autumn, we do 3 sensitivity experiments by switching off the biogenic, lightning, and biomass burning emissions, respectively. The separate contribution of the three sources to tropospheric ozone over Africa is shown in the supplement file. In both NH spring and NH autumn, the influence of biogenic emissions is mainly in the upper troposphere, while the effect of lightning $NO_x$ peaks at lower levels. In NH spring, the contributions of the two sources in NH are comparable to that in SH. However, in NH autumn, the contributions of the two sources are mainly in the SH. Biogenic and lightning emissions are both the important sources for African tropospheric ozone. We have revised our discussion to make it more in-depth.*

10: Section 4.2 It seems that Figure 8 and Figure 9 are already sufficient for the discussion. I suggest to remove Figure 10 to make the paper more concise.

*Thanks for the points. Fig. 10 in the last version not only supports the discussion for Figs. 8 and 9 in the last version but also provides additional information that is not available in these two figures. First, Fig. 10 shows the seasonal variation of the inflow and outflow flux of African ozone over Asia directly indirectly instead of separating the flux into ozone concentrations and winds. Second, the influence of the seasonal variation of the westerly jet is more clearly presented than Figs. 8 and 9. Third, the difference between the inflow and outflow flux is clearly shown, which is not available from other figures. In addition, the influence of the Somali jet on the lower tropospheric over western India in NH summer is captured by the inflow flux in the figure. For these reasons, we think it is better to keep Fig. 10 in this revision (now Fig. 12).*

---

## Author Comment (AC3) · 27 Jan 2018

*We thank the reviewers for their constructive comments, suggestions, and corrections. In this revision, we have addressed the reviewer's concerns. The following is a one-to- one response to their questions.*

*Response to Comments by Anonymous Referee #2*

This is a relatively straightforward analysis of the interplay of meteorological processes and atmospheric chemistry in venting out ozone and ozone precursors from Africa and reaching Africa. The manuscript is well written, the figures appropriate.

As the authors state- there is relatively little literature discussing Africa-to-Asia transport, so this is a welcome addition, despite it doesn't make use of the recommendation in the HTAP2 exercise to harmonize region definitions to allow comparability of results.

*We thank the reviewer for the encouragement and for the valuable and thoughtful comments. In Fig. 1, we show the regional definitions of the HTAP2and in our analysis so the reader can have an idea for the similarity and difference between the two definitions.*

A minor remark is that I don't see terribly much added value of the trajectory analysis in figure 12.

*The trajectory analysis is improved in this revision from the following aspects. (1) Trajectories in two more seasons, spring and autumn, are added for comparison between the four seasons instead of just two seasons in the last version, (2)trajectories from four more stations in Africa are added for a wider coverage and representation, (3)the mean transport pathways for the trajectories that arrive Asia are illustrated with lapse time indicated, and (4) more discussions are added in the revision on the mechanisms that control the transport of African ozone to Asia in section 3.3.*

Although some attempt has been made to demonstrate the model's ability to model ozone over Africa, I think this could be done more convincingly- there is meanwhile a

host of other observations (surface, aircraft, satellite tropospheric ozone columns) that could be explored.

*Thanks for the point. In this revision, we add the validation between ozonesonde observations and GEOS-Chem simulations that shows the performance of the model in simulating the seasonal and interannual variability of tropospheric ozone over Africa in the surface layer, lower, middle, and upper troposphere. The comparison at three more ozonesonde stations in India is added. Furthermore, the ozone data from the Tropospheric Emission Spectrometer (TES) satellite instrument are used to evaluate the GEOS-Chem simulation in the middle troposphere (464 hPa) globally. The detailed comparisons are shown in Figs. 2-4 and Tables1-2 and discussed in section 2.2.*

Are signals from African ozone visible in soundings over India?

*The transport of airmass from African in summer is reflected in the ozonesonde data at Poona and Thiruvananthapuram in western India (in an added figure, Fig. 4). The effect of the Somali jet on western India is obvious as low ozone concentrations appear in the lower troposphere in NH summer.*

The organization and discussion of methods could be somewhat more systematic.

*Thanks for the suggestions. The methods are reorganized and the presentation is polished.*

I suggest that the authors explore somewhat further these aspects, and recommend the manuscript to be accepted after taking these major and minor comments below into account.

Minor comments.

l. 11-30 the abstract could be somewhat more explicit in describing the regions and attribution methodology.

*Thanks. The abstract is rewritten to include the regional description and attribution*

*methods.*

l. 16 Replace boreal by NH winter. Or find better way of describing which months are discussed. Are the > and < really meant to express minima and maxima?

*Thanks. We have replaced boreal winter by northern hemisphere (NH) winter in this revision. We have removed the > and <, and used certain numbers to express minima and maxima.*

l. 30 I miss some statement on the relevance of this analysis. How much of the Asian ozone was produced in Africa or from African precursor emissions- where is it most important (not only vertical but also geographically.

*The sentence has been rephrased for clarity. Vertically, the influence of African ozone is mainly in the middle and upper troposphere. Geographically, the imported African ozone mainly distributes over latitudes south to 40°N in Asia.*

l. 35 give reference time to which this RF estimate pertains.

*Thanks. The reference time is given: "It also acts as a greenhouse gas, whose global mean radiative forcing is about 0.4 ±0.2 W/m² for the period 1750-2011 (Myhre et al., 2013)".*

l. 46 add: as well as a range of papers in the HTAP2 (Galmarini 2016) special issue.

*Thanks. Galmarini et al. (2017) and several other papers in the HTAP2 special issue are now added in the citations.*

l. 53 The issue is also very connected to legislative issues related to the control of ozone and ozone exceedance in the western states of the USA, e.g. as discussed in Huang et al. (2017; already cited).

*Thanks. This point is now included in Introduction.*

l. 54 One reference on LRT transport between South Asia and East Chakraborty et al.

Science of the Total Environment, 523, 2015

*Thanks for your recommendation. The reference is helpful to the study and it is added in the citations.*

l. 73 … makes a contribution … how is contribution defined? Zero out of emissions? This is important because later you present a different method.

*We have clarified this term in Introduction. The contribution of the source regions to the receptor region can presented as absolute or fractional contribution. The former refers to the concentrations of imported ozone in a unit of ppbv, while the latter is the ratio of the imported ozone to the total ozone in a grid, a layer, or a region. We have also rephrased this sentence to make it clear.*

l. 117 which resolution is used for GEOS-CHEM; what was the underlying resolution of the assimilation product. Importantly for this paper, how is convection parameterized, is there any evaluation over Africa of these process. Interhemispheric mixing and similar: refer to any relevant application of the model that demonstrates it is fit-for-purpose for this study. I realize that these are discussed later, but I would have expected these descriptions here.

*Thanks for your comment. The detailed descriptions of GEOS-Chem have been moved to an earlier part section 2.1. GEOS-4 uses the schemes developed by Zhang and McFarlane (1995) for deep convection and by Hack (1994) for shallow convection. GEOS-4 meteorology is found to be characterized with stronger deep convection in tropics than GEOS-5 (Liu et al., 2010; Zhang et al., 2011). Liu et al. (2010) and Zhang et al. (2011) have shown good agreement of GEOS-Chem simulations driven by GEOS-4 with satellite observations in tropical troposphere. Choi et al. (2017) compared the simulations of the Global Modeling Initiative (GMI) chemistry and transport model (CTM) driven by three meteorological data sets (fvGCM for 1995, GEOS-4 for 2005, MERRA for 2005) with ozonesonde and TES observations. They found that ozone simulated by GEOS-4 has the highest correlation with the observations. The information about GEOS-4 is also described in Zhu et al. (2017b).*

*We have not performed direct evaluation on the convection parameterization schemes. Nevertheless, these previous studies and the good validation results over Africa from this study provide us confidence on the model performance. We have revised the manuscript to explain the model's strength and limitations.*

l. 125- If I understand correctly the authors merge the EDGAR3.2 global inventory with regional ones. Which period? How do these inventories compare with e.g. the HTAP2 inventory for 2008/2010 in this special issue, or EDGAR4.2 products (for time series).

*We do the full chemistry simulation in GEOS-Chem to generate ozone production and loss data in 2005 with the emission data from the global EDGAR 3.2 inventory for 2000, the INTEX-B Asia emissions inventory for 2006, the NEI05 inventory in North America for 2005, the EMEP inventory in Europe for 2000, the BRAVO inventory in Mexico for 1999, and the CAC inventory in Canada for 2005. This part has been described more in section 2.1.*

*The comparison of the inventories used in the study with the HTAP2 inventory for 2008 is shown the supplement file for CO and for $NO_x$. Compared to HTAP2 inventory for 2008, the CO emissions used in GEOS-Chem is higher in North America, Europe and East China, and lower in Africa and Southeast Asia throughout the year. The $NO_x$ emissions are higher in North America and Europe, and lower in South Asia.*

*The contribution of anthropogenic emissions to African ozone is smaller than that of other emissions. Aghedo et al. (2007) estimated that anthropogenic emissions emitted in Africa account for approximately 11% (4.7Tg/42.8Tg) of the African emissions influencing the global tropospheric ozone burden. Therefore, the slightly difference in the anthropogenic $NO_x$ and CO emissions over Africa between the two may have little impact on our analysis.*

l. 135 What is the global lightning source strenght and specific for Africa. How does this compare to other studies.

*Lightning $NO_x$ emission used in the study is shown by annual mean and in each*

*season in the supplement file. The annual global lightning $NO_x$ source amount is 5.97*
*Tg N yr$^{-1}$, comparable to 6±2 Tg N yr$^{-1}$ in Martin et al. (2007) and 6.3 Tg N yr$^{-1}$ in*
*Miyazaki et al. (2014). Miyazaki et al. (2014) estimated the annual global lightning*
*$NO_x$ emission by assimilating observations of NO2, HNO3, and CO measured by OMI,*
*MLS, TES, and MOPITT into the global chemical transport model CHASER. The*
*annual lightning emission is 1.72 Tg N month$^{-1}$ in Africa, 0.80 Tg N month$^{-1}$ in NHAF,*
*and 0.79 Tg N month$^{-1}$ in SHAF, shown in the supplement file.*
*Reference: Martin, R. V., Sauvage, B., Folkins, I., Sioris, C. E., Boone, C., Bernath, P.,*
*and Ziemke, J.: Space-based constraints on the production of nitric oxide by lightning,*
*J. Geophys. Res., 112, D09309, doi:10.1029/2006JD007831, 2007.*

l. 137 briefly describe what is the 'standard' tagged ozone method. Pro's and
con's-limitations. Comes now later

*The tagged ozone method tags ozone by the region where ozone is generated. The*
*method was first proposed by Wang et al. (1998) and then developed and used by a*
*number of studies on ozone transport (Fiore et al., 2002; Sudo and Akimoto, 2007;*
*Zhang et al., 2008; Liu et al., 2011; Sekiya and Sudo, 2012, 2014; Zhu et al., 2017b). It*
*is done by the following two steps. First, the production rates and loss frequencies of*
*odd oxygen ($O_x$= $O_3$+$NO_2$+2$NO_3$+3$N_2O_5$+$HNO_3$+$HNO_4$+PAN+PMN+PPN) were*
*generated and archived from a full chemistry simulation before the tagged simulation.*
*Since ozone accounts for most of $O_x$, we refer to ozone instead of $O_x$ for clarity.*
*Second, GEOS-Chem is run again in the tagged ozone mode using the ozone production*
*and loss data, with separate tracers for the ozone produced from each specific source*
*region. The tagged ozone tracer method can assess the contributions of ozone*
*produced in Africa to other regions. The advantages and limitations of the tagged*
*ozone simulation are discussed in section 2.1.*

l. 138 I expected this description earlier.

*Thanks. The description has been moved into an earlier part of the section.*

l. 140- 143: better include with the GEOSCHEM description.

*Thanks. This part has been included with the GEOS-Chem description.*

l. 150- what is the reason for not simply taking the 'african mask'- instead two blocks. I suggest adding a simple figure, showing these masks on top of a map (perhaps along with the HTAP2 definition of Africa).

*Thanks for the suggestion. Figure 1 is added to show the definitions of the source and receptor regions in this study. The definition of Africa in HTAP2 is also presented to show the similarity and difference between the two definitions. The reason for using the blocks to define regions is because this is the way that GEOS-Chem uses for the tagged ozone simulation mode.*

l. 155 I think most papers that I know keep (anthropogenic) emissions constant- but that is not necessarily the same as keeping the production terms constant. What could be the impact?

*In this study, the interannual variation is driven by meteorology only with the fixed daily ozone production rate and loss frequency in 2005. We have generated ozone production rate and lose frequency in other two separate years. Using these two sets of daily ozone production and loss frequency, two additional time series of 20-year simulations can be generated. The three time series show consistent interannual variation in transport of African ozone to the Asian troposphere. This sensitivity test suggests that our results on the interannual variation are robust. The differences in the three sets of ozone production and loss rates reflect partially the differences in emissions in the three years. It is likely that with fixed emissions, the differences in the three sets of ozone production and loss rates will be smaller.*

l. 175- what about sfc observations, tropospheric residual from satellite. The comparison is fairly superficial

*Thanks. In the revision, we have included more comparisons between the GEOS-Chem simulation and ozonesonde observation by adding 2 figures and two*

*tables. The seasonal and interannual variation GEOS-Chem simulations have been further compared with ozonesonde data in the surface layer, lower, middle, and upper troposphere at the African sites. Three ozonesonde stations at India are selected for the comparison. Figs. 2 and 3 and Tables 1 and 2 are added in section 2.2. The correlation coefficient (r), bias in percentage, and root-mean-square error (RMSE) between the simulations and the ozonesonde data are presented. The comparison in global distribution with the TES satellite ozone data are shown in the supplement file.*

l. 201 Imported ozone=>try to describe more exactly what it is. Region-average abundance of imported ozone (imported ozone could be also the flux through the western border, for instance).

*Thanks for the point. We define ozone generated in Africa under the tropopause as African ozone. Following Holloway et al. (2008), "Imported ozone" is used to refer ozone that is distributed over the receptor region. In the paper, when we discuss African ozone over Asia, we use "imported African ozone" to differentiate it from the total ozone concentrations in Asia. We now have clarified this term in Introduction.*

p. 295 /figure 6: I guess if the units are molec/cm2/s this pertains to the integrated amount over a model layer; otherwise it should rather be per cm3?

*Thanks for the point. Yes, in the last version, the unit pertains to the integrated $NO_x$ amount over a model layer. We have converted the unit into $NO_x$ emissions per cubic meter per second. Therefore, the lightning $NO_x$ emissions are expressed as $molec/m^3/s$ in this revision.*

---

## Author Response (AR3)

*We thank the reviewers for their constructive comments, suggestions, and corrections. This version has been largely improved owing to their helpful review. In our revision, we have addressed the reviewer's concerns. The following is a one-to-one response to their questions. Following the reviewers' suggestions, the major improvements in this revision are as follows.*

1. *We have reconstructed the paper to make it more concise. The presentation is polished throughout and more in-depth discussion is added. A large part of the paper is rewritten.*
2. *We have added more details on the validations between the ozonesonde observations and GEOS-Chem simulations.*
3. *More details are provided and discussed about GEOS-Chem, including the emission inventories, the tagged ozone simulations, and other issues raised by the reviewers.*
4. *The trajectory analysis is improved in this revision. More discussions are added in this revision throughout section 3.3, in section 4.2, and in the conclusions (section 5).*

*A manuscript with tracked changes is followed.*
* * *
*Response to Comments by Anonymous Referee #1*

General Comments:

The authors provide a detailed analysis for the long-range transport of tropospheric $O_3$ from Africa to Asia. They indicated that African $O_3$ have important influences on free tropospheric $O_3$ over Asia, and the imported African $O_3$ peaks in winter because of the shifts of transport and emission patterns. I recommend the paper for publication after consideration of the points below.

1) The paper isn't concise enough for me. For example, Section 5 provides a summary for the transport and emission processes, which is actually a repeat of Section 4.2. In addition, considering the small contribution from SHAF (shown by Figure 4), it may not be necessary to have an individual section (Section 4.3) to discuss its influence.

*Thanks for the comments and suggestions. We have reconstructed the paper to make it more concise. Section 5 in the last version has been removed. The presentation is polished throughout the paper and more in-depth discussion is added. We have reconstructed the section on the interhemispheric transport of ozone from SNAF and added more analyses in this section (now section 4.2). Therefore, we think it is better*

*to keep this section.*

2) The discussion should be improved. The authors should explain why the seasonal variability of biogenic isoprene is so weak (Figure 6); and revise the discussion about the contributions from various sources (i.e. biogenic, biomass burning and lightning, Section 4.2).

*Thanks for the points. In the last version, the color scale for the seasonal variation of biogenic isoprene in Figure 6 was not appropriate so that the seasonal variation was not shown apparently. We have edited the color scale so to better show the magnitude of the seasonal variation of biogenic isoprene (now Figure 8). The regional means of the seasonal variation of biogenic isoprene over Africa is shown in Figure 9, which shows large seasonality of biogenic emissions. The discussion about the contributions from various sources has been revised and discussed in more depth (now in section 3.3).*

Specific Comments:

1: Line 147-149 Are the $O_3$ production and loss rates generated using the full-chemistry simulation with the current model settings or from other studies (Wang et al. 1998; Zhang et al. 2008)?

*In this study, we generated the ozone production and loss rates from the full-chemistry simulation using the current model settings of GEOS-Chem v9-02. We have clarified this in this revision (see section 2.1).*

2: Line 149-153 It would be better to show these regions as boxes in the map (e.g. Figure 5). It is difficult to imagine the regions just based on these lat/lon numbers.

*Thanks for your suggestion. We have added Fig. 1 to show the definitions of the regions. The sites used in the GEOS-Chem validation and trajectory analysis are also shown in Fig. 1.*

3: Line 155-157 Did the authors evaluate the possible influences from interannual

variations of meteorology on chemistry?

*Thanks for the question. In this study, we focus on the impact of meteorology on the transport. Therefore, we keep the ozone production and loss rate fixed in one year and allow the meteorology to vary from year to year.*

*Yes, meteorology also affects chemistry. If keeping ozone production term constant, the meteorology influence on chemistry is ignored. We have pointed this out in this revision (section 2.1, the last 2$^{nd}$ paragraph). We have not directly evaluated this impact. Instead, we have tested if this impact will significantly alter our results. Therefore, we conduct a sensitivity test. First, we run GEOS-Chem in full chemistry mode to generate ozone production rate and lose frequency in other two different years. Years 2001 and 2004 are selected because these are years when the extreme anomalies of imported African ozone appear in Asia. In the two full chemistry simulations, we used the same anthropogenic and biomass burning emissions in 2005 but with the different meteorology in 2001 and 2004 respectively. Therefore, the differences in the ozone production rate and loss frequency data can be regarded as the meteorology influences on chemistry as the emissions are fixed. Then we use these two sets of daily ozone production and loss frequency to run the tagged ozone simulations for 20 years. First, we compare the 20-year mean of imported African ozone from these two runs with the default run in Fig.1. The differences of imported African ozone over Asia between three datasets are small, varying from -1 ppbv to 0.2 ppbv for most layers and months, although the differences are large in NH winter over the upper troposphere, reaching 1-3 ppbv.*

*Second, we compare the three datasets on the interannual variations of imported African ozone over Asia in Fig. 2. Indeed, there are differences between the three datasets. However, the interannual variations of the three simulations are similar and all positive correlated to the ITCZ. This sensitivity test suggests that our treatment is robust in capturing the variation of ozone transport from Africa to Asia from year to year.*

[Figure]

*Figure S1. The differences of imported African ozone over Asia between the 2005 and the 2001 datasets (1st col.) and between the 2005 and the 2004 dataset (2nd col.). The values are averaged over 60-145oE from 1987 to 2006. Associated with the three datasets, the ozone production and loss data are generated with the same emissions but different meteorology in 2001, 2004, and 2005.*

[Figure]

*Figure 2. The interannual variations of the anomaly of imported African ozone over Asia simulated using the three sets of ozone production and loss data at (a) 200 hPa, (b) 600 hPa, and (c) 975 hPa in January. Associated with the three datasets, the ozone production and loss data are generated with the same emissions but different meteorology in 2001, 2004, and 2005.*

4: Line 166-167 Is there any other station available? Why are these three stations selected?

*GEOS-Chem simulations have been validated with ozonesonde data extensively, for example, in North America (Zhang et al., 2008; Zhu et al., 2017b), Europe (Kim et al., 2015), and East Asia (Wang et al., 2011; Zhu et al., 2017a; Zhu et al., 2017b). However, few studies have validated the simulations in Africa. We specifically validate the performance of GEOS-Chem over Africa for an enhanced confidence on our analysis. These three stations are selected for their representative locations and relative long records in Africa. In this version, the ozone data from three ozonesonde stations in India are added to compare with the GEOS-Chem simulations. In addition, the Tropospheric Emission Spectrometer (TES) satellite observations are compared with the GEOS-Chem simulation in the middle troposphere shown in the supplementary file (Fig. S6).*

5: Section 2.3 Is the meteorological data the same as used by the HYSPLIT model? If they are the same, it would be better to combine Section 2.2 with Section 2.3.

*The meteorological data are the same as what are used by the HYSPLIT model. The two parts have been combined into Section 2.3.*

6: Line 237-239 The influence of African $O_3$ to south America across Atlantic is discussed, but isn't shown in the Figure. It could be better to remove the discussion about the transatlantic transport here.

*The discussion about the transatlantic transport here has been removed.*

7: Line 262-269 Although may not be necessary to explain, I am just curious about the reason for the discrepancy between western and eastern Africa.

*In general, the latitudinal position of ITCZ follows the rotation of the sun. In eastern Africa, the seasonal migration of ITCZ with latitude is more symmetrical around the equator, while in western Africa, the migration is limited (Collier and Hughes, 2011). The seasonal migration of the ITCZ in western Africa is complicated. Generally, in*

*NH summer, the convergence zone is formed by the flows from the Atlantic cold tongue and the Saharan heat low, locating around 20ºN (Nicholson, 2009, 2013). In NH winter, the anticyclonic wind from northern Africa converges with the southerly wind from Atlantic. The ITCZ over western Africa still stays in the continent (Nicholson, 2013). Therefore, the seasonal migration of the ITCZ in western Africa is within a narrower range of latitudes than in eastern Africa.*

8: Line 276-277 Figure 6 shows significant seasonal variation for biomass burning CO. Surprisingly, the seasonal variation of biogenic isoprene is ignorable, which seems inconsistent with other study (e.g. Marais et al. 2014). Is it associated with the color scale? On the other hand, the normalized magnitudes of seasonal variability (Figure 7) are comparable between CO and isoprene. Is it due to the usage of standard deviation in the calculation? The approach for normalization is confusing.

Marais, E. A., Jacob, D. J., Guenther, A., Chance, K., Kurosu, T. P., Murphy, J. G., Reeves, C. E., and Pye, H. O. T.: Improved model of isoprene emissions in Africa using Ozone Monitoring Instrument (OMI) satellite observations of formaldehyde: implications for oxidants and particulate matter, Atmos. Chem. Phys., 14, 7693-7703, https://doi.org/10.5194/acp-14-7693-2014, 2014.

*Thanks for the points. Yes, the narrow seasonal variation of biogenic isoprene shown in the old Fig. 6 indeed is due to the use of the color scale. We have edited the color scale in the figure (now Fig. 8) so that the seasonal variation of biogenic isoprene in Africa is better presented. Fig. 9 (old Fig. 7) shows the seasonal variability of isoprene and CO. The biogenic emission peaks in spring and autumn. The magnitude of biogenic isoprene is comparable to the results in Marais et al. (2014). Normalization is not taken in this revision to avoid confusing.*

9: Line 315-362 The discussion in this section is superficial. The authors discuss the contributions from various sources without detailed calculations. For example, the authors indicated: 1) "In boreal spring, a region with high ozone concentrations (>40 ppbv) appears in higher altitudes and … mainly due to the highest biogenic emissions

in the NHAF" 2) "In boreal autumn, the locations of the ITCZ and the Hadley cell are similar to these in boreal spring. Ozone in the African middle troposphere ... attributed to stronger lightning NOx emission" However, there is no evidence to demonstrate that the contributions from biogenic and lightning activities are evaluated carefully. The discussion is simply based on the spatial distribution of Figure 6. The biogenic and lightning activities are highly similar between spring and fall, and it is hard to explain why the spring-time $O_3$ is biogenic dominant, whereas autumn-time $O_3$ is lightning dominant.

*Thanks for the comments. Aghedo et al. (2007) has suggested that the biogenic and lightning emissions are the two important sources influencing African middle and upper tropospheric ozone and affecting global tropospheric ozone burden. To further explore the differences between the situations in NH spring and in NH autumn, we have conducted 3 sensitivity experiments by switching off the biogenic, lightning, and biomass burning emissions, respectively. The separate contribution of the three sources to tropospheric ozone over Africa is shown in the supplementary file (Fig. S7). In HN spring and autumn, the influence of ozone from SHAF on Asia is small and similar (Fig. 7) so we can mainly focus on ozone in NHAF. It appears that in NH spring, elevated ozone abundances from biogenic emissions are higher than that from lightning $NO_x$ emissions while in NH autumn, elevated ozone abundances from biogenic emissions are lower than that from lightning $NO_x$ emissions (Fig. S7, and also see Aghedo et al. (2007)). We have revised our paper accordingly. We also have made the discussion in more depth in section 3.3.4.*

10: Section 4.2 It seems that Figure 8 and Figure 9 are already sufficient for the discussion. I suggest to remove Figure 10 to make the paper more concise.

*Thanks for the points. Fig. 10 (in the last version) has been removed.*

*Response to Comments by Anonymous Referee #2*

This is a relatively straightforward analysis of the interplay of meteorological processes and atmospheric chemistry in venting out ozone and ozone precursors from Africa and reaching Africa. The manuscript is well written, the figures appropriate.

As the authors state- there is relatively little literature discussing Africa-to-Asia transport, so this is a welcome addition, despite it doesn't make use of the recommendation in the HTAP2 exercise to harmonize region definitions to allow comparability of results.

*We thank the reviewer for the encouragement and for the valuable and thoughtful comments. In Fig. 1, we show the regional definitions in this study and in HTAP2, so the reader can have an idea for the similarity and difference between the two definitions.*

A minor remark is that I don't see terribly much added value of the trajectory analysis in figure 12.

*The trajectory analysis is improved in this revision as follows. (1) Trajectories in two more seasons, spring and autumn, are added for comparison between the four seasons instead of just between two seasons in the last version, (2) trajectories from four more stations in Africa are added for a wider coverage and representation, (3) the mean transport paths for the trajectories that arrive Asia are illustrated with lapse times indicated, and (4) more discussions are added in this revision to supplement the discussion on the mechanisms that control the transport of African ozone to Asia throughout section 3.3, in section 4.2, and in the conclusions (section 5).*

Although some attempt has been made to demonstrate the model's ability to model ozone over Africa, I think this could be done more convincingly- there is meanwhile a host of other observations (surface, aircraft, satellite tropospheric ozone columns) that could be explored.

*Thanks for the point. In this revision, we add more validations between the*

*ozonesonde observations and GEOS-Chem simulations from perspectives of variceal profiles and the seasonal and interannual variability of tropospheric ozone over Africa in the surface layer, lower, middle, and upper troposphere. The comparisons at three more ozonesonde stations in India are added. Furthermore, the ozone data from the Tropospheric Emission Spectrometer (TES) satellite instrument are used to evaluate the GEOS-Chem simulation in the middle troposphere (464 hPa) globally in the supplementary file (Fig. S6). The detailed comparisons are shown in Figs. 2-4 and Tables1-2 and are discussed in section 2.2.*

Are signals from African ozone visible in soundings over India?

*The transport of airmass from African in summer is reflected in the ozonesonde data at Poona and Thiruvananthapuram in western India (in an added figure, Fig. 4). The impact of the Somali jet on the ozone in western India is obvious as low ozone concentrations appear in the lower troposphere in NH summer.*

The organization and discussion of methods could be somewhat more systematic.

*Thanks for the suggestions. The methods are reorganized and the presentation is polished in section 2.*

I suggest that the authors explore somewhat further these aspects, and recommend the manuscript to be accepted after taking these major and minor comments below into account.

*Thanks. We have followed these suggestions.*

Minor comments.

l. 11-30 the abstract could be somewhat more explicit in describing the regions and attribution methodology.

*Thanks. The abstract is rewritten to explicitly state the regional definitions and attribution methods.*

l. 16 Replace boreal by NH winter. Or find better way of describing which months are discussed. Are the > and < really meant to express minima and maxima?

*Thanks. We have replaced boreal winter with northern hemisphere (NH) winter in this revision. We have removed the > and <, and used certain numbers to express minima and maxima.*

l. 30 I miss some statement on the relevance of this analysis. How much of the Asian ozone was produced in Africa or from African precursor emissions- where is it most important (not only vertical but also geographically.

*Vertically, the influence of African ozone on Asia is mainly in the middle and upper troposphere. Geographically, the imported African ozone mainly distributes over latitudes south to $40^o N$ in Asia. We have added more detail about this in the abstract.*

l. 35 give reference time to which this RF estimate pertains.

*Thanks. The reference time is given: "It also acts as a greenhouse gas, whose global mean radiative forcing is about $0.4 \pm 0.2$ W/m$^2$ for the period 1750-2011 (Myhre et al., 2013)".*

l. 46 add: as well as a range of papers in the HTAP2 (Galmarini 2016) special issue.

*Thanks. Galmarini et al. (2017) and several other papers in the HTAP2 special issue are now added in the citations.*

l. 53 The issue is also very connected to legislative issues related to the control of ozone and ozone exceedance in the western states of the USA, e.g. as discussed in Huang et al. (2017; already cited).

*Thanks. This point is now included in Introduction.*

l. 54 One reference on LRT transport between South Asia and East Chakraborty et al. Science of the Total Environment, 523, 2015

*Thanks for this recommendation. The reference is helpful to the study and it is added*

*in the reference.*

l. 73 … makes a contribution … how is contribution defined? Zero out of emissions? This is important because later you present a different method.

*We have clarified this term in Introduction. In this study, we used the tagged ozone simulation to track ozone from source regions to a receptor region. We did not use the perturbation simulations that turn the emissions off to see the contribution of source regions to a receptor region.*

*The contribution of the source regions to the receptor region can be presented as absolute and fractional contributions. The former refers to the concentrations of the imported ozone in a unit of ppbv, while the latter is the ratio of the imported ozone to the total ozone in a grid, a layer, or a region. We have also described the term in Introduction and rephrased this sentence to make it clear.*

l. 117 which resolution is used for GEOS-CHEM; what was the underlying resolution of the assimilation product. Importantly for this paper, how is convection parameterized, is there any evaluation over Africa of these process. Interhemispheric mixing and similar: refer to any relevant application of the model that demonstrates it is fit-for-purpose for this study. I realize that these are discussed later, but I would have expected these descriptions here.

*Thanks for these comments. The detailed descriptions of GEOS-Chem have been moved to an earlier part in section 2.1. In this study, the simulations are driven by GEOS-4 meteorology at a $4^o$ latitude by $5^o$ longitude horizontal resolution, degraded from their native resolution of $1^\circ$ latitude $\times 1.25^\circ$ longitude. There are 30 vertical layers including 17 levels in the troposphere (see the $2^{nd}$ paragraph in section 2.1). GEOS-4 uses the schemes developed by Zhang and McFarlane (1995) for deep convection and by Hack (1994) for shallow convection. GEOS-4 meteorology is found to be characterized with stronger deep convection in tropics than GEOS-5 (Liu et al., 2010; Zhang et al., 2011). Liu et al. (2010) and Zhang et al. (2011) have shown good agreement of GEOS-Chem simulations driven by GEOS-4 with satellite observations*

*in the tropical troposphere. Choi et al. (2017) compared the simulations of the Global Modeling Initiative (GMI) chemistry and transport model (CTM) driven by three meteorological data sets (fvGCM for 1995, GEOS-4 for 2005, MERRA for 2005) with ozonesonde and TES observations. They found that ozone simulated by GEOS-4 has the highest correlation with the observations. These previous studies and the good validation results over Africa from this study provide us confidence on the model performance. We have provided more details on the model descriptions in this revision (see section 2.1).*

l. 125- If I understand correctly the authors merge the EDGAR3.2 global inventory with regional ones. Which period? How do these inventories compare with e.g. the HTAP2 inventory for 2008/2010 in this special issue, or EDGAR4.2 products (for time series).

*We conducted the full chemistry simulation in GEOS-Chem to generate ozone production and loss data in 2005. The merged emission data are from the global EDGAR 3.2 inventory in the base year 2000. The regional emission inventories include the INTEX-B Asia emissions inventory in 2005 with base year 2006, the NEI05 inventory in North America in the base year 2005, the EMEP inventory in Europe in the base year 2005, the BRAVO inventory in Mexico in 2005 with base year 1999, and the CAC inventory in Canada in the base year 2005. This part has been described in more details in section 2.1.*

*We compared the anthropogenic emissions of CO and $NO_x$ from the GEOS-Chem for 2000 inventories with those in the HTAP2 inventories for 2008 (http://edgar.jrc.ec.europa.eu/htap_v2/) and showed the comparison in the supplementary file (Figs. S3 and S4). Compared to the HTAP2 inventory for 2008, the CO and $NO_x$ emissions in GEOS-Chem are lower in Africa throughout the year. The annual anthropogenic emissions of CO in Africa in GEOS-Chem and from the HTAP2 inventory are about 12.2 Tg $yr^{-1}$ and 62.5 Tg $yr^{-1}$, respectively. The anthropogenic emissions for NOx in Africa in GEOS-Chem and HTAP are about 2.27 Tg $yr^{-1}$ and 4.53 Tg $yr^{-1,}$ respectively*

*The contribution of anthropogenic emissions to generation of African ozone is considered to be smaller than that of other emissions. Aghedo et al. (2007) estimated that anthropogenic emissions emitted in Africa account for approximately 11% (4.7Tg/42.8Tg) of the total African emissions that impact the global tropospheric ozone burden. In this study, the annual biomass emissions of CO in Africa in GEOS-Chem are 182.8 Tg yr$^{-1}$, which is much larger than anthropogenic emissions of CO (4.7Tg yr$^{-1}$). Nevertheless, we should consider the impact of this issue and we have stated in this revision that "Although the anthropogenic emissions contribute less significantly to the ozone generation in Africa than biogenic, biomass burning, and lightning emissions (Aghedo et al., 2007), the differences between these emission inventories imply that African ozone simulated by GEOS-Chem is with some uncertainties." (Line 187-190). .*

l. 135 What is the global lightning source strenght and specific for Africa. How does this compare to other studies.

*Lightning NO$_x$ emission used in the study is shown by annual mean and in each season in the supplementary file (Fig. S1). The annual global lightning NO$_x$ source amount is 5.97 Tg N yr$^{-1}$, comparable to 6±2 Tg N yr$^{-1}$ in Martin et al. (2007) and 6.3 Tg N yr$^{-1}$ in Miyazaki et al. (2014). Miyazaki et al. (2014) estimated the annual global lightning NO$_x$ emission by assimilating observations of NO$_2$, HNO$_3$, and CO measured by OMI, MLS, TES, and MOPITT into the global chemical transport model CHASER. The annual lightning emission is 1.72 Tg N month$^{-1}$ in Africa, 0.80 Tg N month$^{-1}$ in NHAF, and 0.79 Tg N month$^{-1}$ in SHAF, shown in the supplementary file. We have added the information in section 2.1 in the 3$^{rd}$ paragraph.*

l. 137 briefly describe what is the 'standard' tagged ozone method. Pro's and con's-limitations. Comes now later

*The tagged ozone method tracks ozone that is generated in different regions. The method was first proposed by Wang et al. (1998) and then further developed and used by a number of studies (Fiore et al., 2002; Sudo and Akimoto, 2007; Zhang et al., 2008;*

*Liu et al., 2011; Sekiya and Sudo, 2012, 2014; Zhu et al., 2017b). This is one of the standard modes in GEOS-Chem. It is done by the following two steps. First, the daily production rates and loss frequencies of odd oxygen ($O_x=$ $O_3+NO_2+2NO_3+3N_2O_5+HNO_3+HNO_4+PAN+PMN+PPN$) were generated and archived from a full chemistry simulation before the tagged ozone simulation. Since ozone accounts for most of $O_x$, ozone instead of $O_x$ is used for clarity. Second, GEOS-Chem is run again in the tagged ozone mode using the archived ozone production rate and loss frequency, with separate tracers for the ozone produced from each of the specific source regions. Therefore, the tagged ozone tracer method can assess the contributions of African ozone to Asia. The advantages of and issues with the tagged ozone simulation are discussed in section 2.1 (see the last paragraph in section 2.1) with an additional figure in the supplementary file (Fig. S5).*

l. 138 I expected this description earlier.

*Thanks. The description has been moved into an earlier part of the section. Please see the 2[nd] paragraph in section 2.1.*

l. 140- 143: better include with the GEOSCHEM description.

*Thanks. This part has been included with the GEOS-Chem description. Please see the 2[nd] paragraph in section 2.1.*

l. 150- what is the reason for not simply taking the 'african mask'- instead two blocks. I suggest adding a simple figure, showing these masks on top of a map (perhaps along with the HTAP2 definition of Africa).

*Thanks for the suggestion. Fig. 1 is added to show the definitions of the source and receptor regions in this study. The definition of Africa in HTAP2 is also presented to show the similarity and difference between the two definitions. The reason for using the blocks to define regions is because this is the default way that GEOS-Chem uses for the tagged ozone simulation.*

l. 155 I think most papers that I know keep (anthropogenic) emissions constant- but that is not necessarily the same as keeping the production terms constant. What could be the impact?

*Thanks for this comment. Meteorology affects both chemistry and transport, a physical process. If keeping ozone production term constant, the meteorology influence on chemistry is ignored. If keeping emissions constant, the impacts of meteorology on both chemistry and transport are considered. We have pointed this out in this revision (section 2.1, the last 2nd paragraph).*

l. 175- what about sfc observations, tropospheric residual from satellite. The comparison is fairly superficial

*Thanks. In this revision, we have included more comparisons between the GEOS-Chem simulations and ozonesonde observations by adding 2 figures and two tables. The seasonal and interannual variation GEOS-Chem simulations have been further compared with ozonesonde data in the surface layer, lower, middle, and upper troposphere at the African sites. Three ozonesonde stations at India are added for the comparison. Figs. 2 and 3 and Tables 1 and 2 are added in section 2.2. The correlation coefficient (r), bias in percentage, and root-mean-square error (RMSE) between the simulations and the ozonesonde data are presented. The comparison in global distribution with the TES satellite ozone data are shown in the supplementary file (Fig. S6).*

l. 201 Imported ozone=>try to describe more exactly what it is. Region-average abundance of imported ozone (imported ozone could be also the flux through the western border, for instance).

*Thanks for the point. We define ozone that is generated over Africa under the tropopause as African ozone. Following Holloway et al. (2008), "Imported ozone" is used to refer ozone that is distributed over the receptor region. In the paper, when we discuss African ozone over Asia, we use "imported African ozone" to differentiate it from the overall ozone concentrations in Asia. We have clarified this term in*

*Introduction (the last paragraph in section 1).*

p. 295 /figure 6: I guess if the units are molec/cm2/s this pertains to the integrated amount over a model layer; otherwise it should rather be per cm3?

*Thanks for the point. Yes, in the last version, the unit pertains to the integrated $NO_x$ amount over a model layer. We have converted the unit into $NO_x$ emissions per cubic meter per second. Therefore, the lightning $NO_x$ emissions are expressed as molec/m$^3$/s in this revision (see Figs. 8 and 9).*

**Characteristics of intercontinental transport of tropospheric ozone from Africa to Asia**

Han Han[1], Jane Liu[1,2], Huiling Yuan[1], Bingliang Zhuang[1], Ye Zhu[1,3], Yue Wu[1], Yuhan Yan[4], Aijun Ding[1]

[1]School of Atmospheric Sciences, Nanjing University, Nanjing, China

[2]Department of Geography and Planning, University of Toronto, Toronto, Canada

[3]Shanghai Public Meteorological Service Centre, Shanghai, China

[4]Chinese Academy of Science, Institute of Atmospheric Physics, Beijing, China

*Correspondence to:* Jane Liu (janejj.liu@utoronto.ca)

**Abstract.**
In this study, we characterize the transport of ozone from Africa to Asia through the analysis of the simulations of a global chemical transport model, GEOS-Chem, from 1987 to 2006. The receptor region Asia is defined within $5^{o}N$-$60^{o}N$ and $60^{o}E$-$145^{o}E$, while the source region Africa is within $35^{o}S$-$15^{o}N$ and $20^{o}W$-$55^{o}E$ and within $15^{
[revised manuscript text omitted]